# Structural insights into the elevator-type transport mechanism of a bacterial ZIP metal transporter

Yao Zhang [1], Yuhan Jiang[2,5], Kaifu Gao[3,5], Dexin Sui[1], Peixuan Yu[1], Min Su [4], Guo-Wei Wei [1,3] & Jian Hu [1,2] ✉

The Zrt-/Irt-like protein (ZIP) family consists of ubiquitously expressed divalent metal transporters critically involved in maintaining systemic and cellular homeostasis of zinc, iron, and manganese. Here, we present a study on a prokaryotic ZIP from *Bordetella bronchiseptica* (BbZIP) by combining structural biology, evolutionary covariance, computational modeling, and a variety of biochemical assays to tackle the issue of the transport mechanism which has not been established for the ZIP family. The apo state structure in an inward-facing conformation revealed a disassembled transport site, altered inter-helical interactions, and importantly, a rigid body movement of a 4-transmembrane helix (TM) bundle relative to the other TMs. The computationally generated and biochemically validated outward-facing conformation model revealed a slide of the 4-TM bundle, which carries the transport site(s), by approximately 8 Å toward the extracellular side against the static TMs which mediate dimerization. These findings allow us to conclude that BbZIP is an elevator-type transporter.

Some *d*-block metals broadly and critically participate in catalysis, macromolecule stabilization, gene regulation, and cell signaling[1–5]. To maximize the benefits while minimize the potential damages of these highly reactive elements, living organisms have evolved dedicated mechanisms to maintain metal homeostasis at systemic and cellular levels, in which the transporters play pivotal roles by controlling the fluxes across plasma membrane and organellar membranes. One prominent example is the Zrt-/Irt-like protein (ZIP) family (or solute carrier family 39, SLC39a)[6–8]. The ZIPs import metal ions, including $Zn^{2+}$, $Fe^{2+}$, $Mn^{2+}$, and other divalent metals in the first row of the *d*-block region in the periodic table, to cytoplasm[6]. Cadmium transport activity of some family members makes them responsible for cadmium uptake and accumulation in plants and animals[9–11]. As an ancient and highly diverse family present in all the kingdoms of life, the ZIP family can be divided into at least four major subfamilies where the eukaryotic ZIPs appear to evolve from distinct prokaryotic ancestors in different

subgroups[7]. In humans, the key functions of a total of fourteen ZIPs have been manifested by inherited diseases caused by loss-of-function mutations[12–18] and aberrant upregulation or downregulation in a variety of cancers[19–26], as well as their involvement in many other diseases[27–31].

Despite of vital biological functions, there are still fundamentally important questions to be answered for the ZIP family, one of which is the transport mechanism. Although the electrochemical gradient of zinc ions across the plasma membrane is believed to allow a passive influx of zinc ions, it has been shown that the ZIP-mediated metal transport may be coupled with co-transportation of other solutes, including bicarbonate for ZIP8 and ZIP14[32,33], and proton for ZIP4[34]. While the kinetic studies using the cell-based transport assays suggested a carrier mode for eukaryotic and prokaryotic ZIPs[35–42], a bacterial ZIP from *Bordetella bronchiseptica* (ZIPB or BbZIP) reconstituted in proteoliposome was reported to behave like an ion channel[43]. The

[1]Department of Biochemistry and Molecular Biology, Michigan State University, East Lansing, MI, USA. [2]Department of Chemistry, Michigan State University, East Lansing, MI, USA. [3]Department of Mathematics, Michigan State University, East Lansing, MI, USA. [4]Life Sciences Institute, University of Michigan, Ann Arbor, MI, USA. [5]These authors contributed equally: Yuhan Jiang, Kaifu Gao ✉e-mail: hujian1@msu.edu

studies on other bacterial ZIPs reconstituted in proteoliposome, however, showed Michaelis-Menten type kinetics[44,45]. Structural studies have provided critical insights into the transport mechanisms for numerous membrane transport proteins. The crystal structure of BbZIP with bound metal substrate revealed an inward-facing conformation (IFC) where the transport site is fully exposed to the cytoplasmic side while the pathway toward the extracellular side is blocked by multiple hydrophobic residues along a distance of approximately 8 Å[7,46]. The analysis of a hydroxyl radical labeling experiment suggested that the blocked pathway may be opened via significant dynamics of the transporter so that the metal substrate is able to enter the transport site from the extracellular side[47]. So far, structures describing alternative conformations of ZIPs have yet been reported.

In this work, we aim to clarify the transport mechanism of BbZIP through a combination of structural biology, analysis of evolutionary covariance, computational modeling, and biochemical assays. The findings, in particular the structure of BbZIP in the apo state and a biochemically validated outward-facing conformation model, consistently support the conclusion that BbZIP is an elevator-type transporter where a four-transmembrane helix (TM) bundle with bound metal substrate slides as a rigid body against the dimeric domain composed of the other static TMs to implement the alternating access mechanism.

## Results

### Crystal structure of BbZIP in the apo state

It is known that substrate binding/release may trigger conformational changes of a membrane transporter by altering the equilibrium between the conformational states. While we were trying to crystallize BbZIP in the apo state to reveal alternative conformations, we found that the EDTA-treated protein was prone to form aggregates. During the extensive crystallization trials in lipidic cubic phase, we identified a couple of low pH conditions (~pH 4.0) where diffracting crystals were reproducibly obtained and found to be in the apo state (see below). Optimization of these conditions led to structure determination of the full length BbZIP with an N-terminal His-tag at the resolution of 2.75 Å by using the previously solved BbZIP structure (PDB entry ID: 5TSB) as the search model in molecular replacement (Supplementary Table 1).

The improved data quality enabled us to build the structural model covering nearly the whole amino acid sequence of BbZIP, except for the first four residues at the N terminus (residues 1-4) and ten residues in the flexible loop connecting α3 and α4 (residues 149-158). An unexpected finding is that BbZIP has nine authentic TMs where the extra TM (referred to as α0 hereafter) is located upstream of the eight TMs (α1-α8) conserved in the entire ZIP family (Fig. 1). α0 appears to weakly associate with α3 and α6 via smooth surfaces consisting of glycine residues. The significantly changed structure of the eight-TM core (discussed later) may better stabilize this otherwise

highly flexible segment. Sequence analysis showed that although α0 is not rare in the gufA subfamily, many gufA subfamily members do not possess it (Supplementary Fig. 1)[7]. A short amphipathic helix (α0a) was also solved at the very N-terminus of BbZIP, which is even less conserved and seems to be stabilized by the interactions with the cytoplasmic extensions of α4 and α3. The functions of α0 and α0a are unknown. As they are located at the peripheral region of the structure and there are only limited interactions with the much more conserved core region, α0 or α0a is unlikely to be essential for transport. With the improved density map, we were also able to build the structural model of the previously missed loop connecting α7 and α8 (resides 278-288) (Supplementary Fig. 2), where a conserved proline residue (P279) causes a kink of α7 at the cytoplasmic side.

Inspection of the electron density at the transport site indicated that there was no bound metal (Fig. 2a), confirming that the protein was crystallized in the apo state. When compared with the metal-bound structure, the binuclear metal center (BMC, the transport sites) underwent large structural changes (Fig. 2b). The eight residues forming the BMC can be divided into three groups based on the extent of structural changes: For N178 from α4, D208 from α5, and E240 from α6, no significant changes were observed; For E181 from α4, and Q207 and E211 from α5, the side chains oriented differently from the metal-bound state, all turning away from the M1 metal binding site; For H177 from α4 and M99 from α2, not only their side chains but also the $C_\alpha$s moved away from the M1 site, making it impossible for them to chelate metal ions with other residues in the BMC. The overall outcome of these structural changes is that the M1 site, which is the primary transport site[48], was completely disassembled whereas the M2 metal binding site was affected to a much lesser extent, even though E181, which is the bridging residue of the BMC, is unlikely to bridge the two metal sites any more in the apo state structure. Two ordered water molecules observed at the BMC may play a role in stabilizing the structure, but since they are located at the positions different from the M1 or M2 site, they cannot maintain the entire geometry of the BMC.

The shift of $C_\alpha$s of H177 and M99 are indicative of structural changes in a scale greater than local conformational rearrangement. The apo state structure is in an IFC but with a root-mean-square deviation (RMSD) of 1.63 Å for $C_\alpha$s when compared with the previously reported cadmium bound structure (PDB entry ID: 5TSB) (Fig. 3a). Further structural inspection revealed that, the RMSD for the helix bundle containing α1/4/5/6 (Domain I hereafter) is 0.56 Å, and that for the other four TMs (Domain II) is 0.79 Å, both of which are significantly smaller than the overall RMSD. This result suggests that the greater overall RMSD likely stemmed from the different orientations for the two domains in these structures. Indeed, when Domain II in the apo state structure was aligned with the corresponding region in the cadmium bound structure, a significant displacement of Domain I was revealed. This structural rearrangement is largely attributed to a

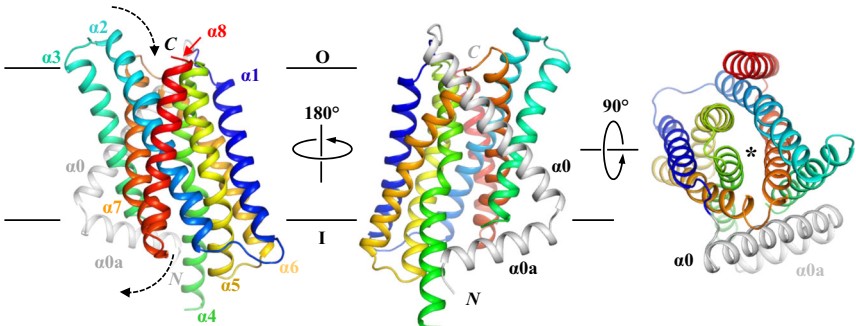

**Fig. 1 | Overall structure of BbZIP in the apo state.** The structures in front view (*left*), back view (*middle*), and top view (*right*) are shown in cartoon mode and colored in rainbow. The extra N-terminal structural elements, including a transmembrane helix (α0) and an amphipathic helix (α0a), are colored in gray. The dashed arrows indicate the entrance and the exit of the transport pathway, which is marked by an asterisk in the top view.

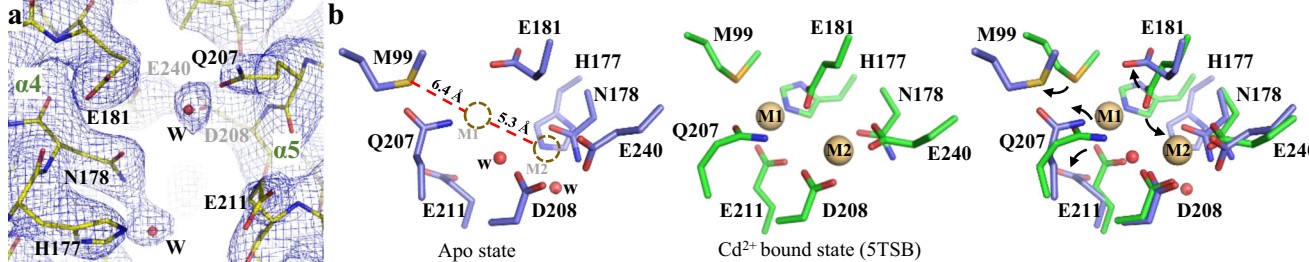

**Fig. 2 | Structure of the BMC of BbZIP in the apo state. a** Electron density map of the BMC (transport sites) of BbZIP in the apo state. The 2FoFc map (σ = 1.2) is depicted as blue meshes. **b** Structural comparison of the BMC in the apo state and the substrate bound state. In the apo state structure, the M1 and M2 sites are marked by dashed circles, and the distances between the M1 site and two chelating residues (M99 and H177) are indicated. The black arrows in the overlapped structure indicate structural rearrangement of the indicated residues. The residues are labeled and shown in stick mode. $Cd^{2+}$ (M1 and M2) and water molecules in the BMC are depicted as spheres in light brown and red, respectively.

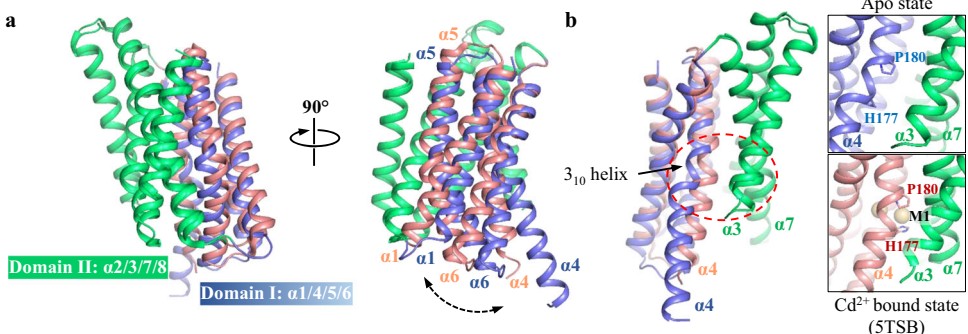

**Fig. 3 | Two-domain architecture revealed by structural comparison between the apo state and the substrate bound state of BbZIP. a** Structural alignment via Domain II. A rigid-body rotation of Domain I (blue for the apo state and pink for the substrate bound state) against Domain II was unraveled. For clarity, α0 and α0a are not shown. **b** Metal removal caused swing of α4 (Domain I) away from α3 and α7 (Domain II). The zoomed-in views of the region in the dashed red oval are shown on the right. H177 and P180 are labeled and depicted in stick mode. The cadmium ions are shown as light brown spheres.

9-degree rigid body rotation of Domain I relative to Domain II with the hinge point (of the rotation) located at the extracellular end of α6. This suggests that Domain I and Domain II interact only through a greasy interface which allows Domain I to slide against Domain II. While this relative movement between the two domains accounted for the large shift of $C_\alpha$ of M99 (Fig. 2b), the drift of H177 was further facilitated by the distortion of α4. Because of the helix-breaking residue P180, α4 is bent at H177 in the cadmium bound structure. In the apo state, the lack of metal ion at the M1 site eliminated a key H177-mediated interaction between α4 and the other TMs in Domain I, leading to the formation of a short $3_{10}$-helix (spanning residues 176-179) at the bending point of α4. As a result, the cytoplasmic side of α4 underwent a large displacement away from α3 and α7 in Domain II (Fig. 3b), which likely facilitates the relative movement between the two structurally independent domains.

Collectively, the apo state structure reveals a two-domain architecture. The disassemble of the M1 site triggered by metal release likely weakens the interdomain interactions, facilitating a rigid body movement of Domain I relative to Domain II via the greasy domain interface.

## Evolutionary covariance analysis

To better analyze and compare the interactions among the TMs, we conducted an evolutionary covariance analysis using EVcouplings [https://v2.evcouplings.org/] and the predicted contacting residues from different transmembrane helices with the probability greater than 75% are listed in Supplementary Table 2. We focused on the predicted interactions between different TMs and mapped the inter-helical interactions on the cross-section view of the BbZIP structure (Fig. 4a). Notably, the same two domains revealed in the structural analysis (Fig. 3a) also emerged in this evolutionary covariance analysis,

where the number of predicted interactions within Domain I or Domain II are much greater than that between them. As a rigid structural unit is stabilized by extensive intradomain interactions, the lack of evolutionarily conserved interactions between Domain I and Domain II implies a weak interaction and thus a mobile interface between them.

Further analysis showed that some of the predicted interactions are not consistent with the solved structures. A portion of these residue pairs are involved in interdomain interactions. For example, out of the six interactions between α2 (Domain II) and α5 (Domain I), four of them, including A95-A218, M99-P210, A102-Q207, and S106-Q207, are 11-15 Å apart (for $C_\alpha s$) in the IFC structures (in the apo or cadmium bound state), excluding the possibility of direct interaction between the involved residues (Fig. 4b). This discrepancy inspired us to explore the possibility that there might be alternative conformations where these predicted interactions are allowed[49]. The results of these efforts are elaborated in later sections. The other predicted contacting residues that are too far apart to interact in the currently available structures are exclusively within Domain II, including those between α3 and α8 and between α7 and α8 (interactions in red in Fig. 4a). As the involved residues are peripheral and pointing away from the core, there is a possibility that they are interacting with the residues of the other protomer when BbZIP form a homodimer via Domain II[50]. The same dimerization interface has been predicted for human ZIP4 in a computational study[51]. In the next section, we experimentally characterized the dimerization interface of BbZIP.

## Dimerization of BbZIP

Accumulated evidence support homo- and heterodimerization for a panel of ZIPs[43,52–55]. For BbZIP, the purified protein in detergent was

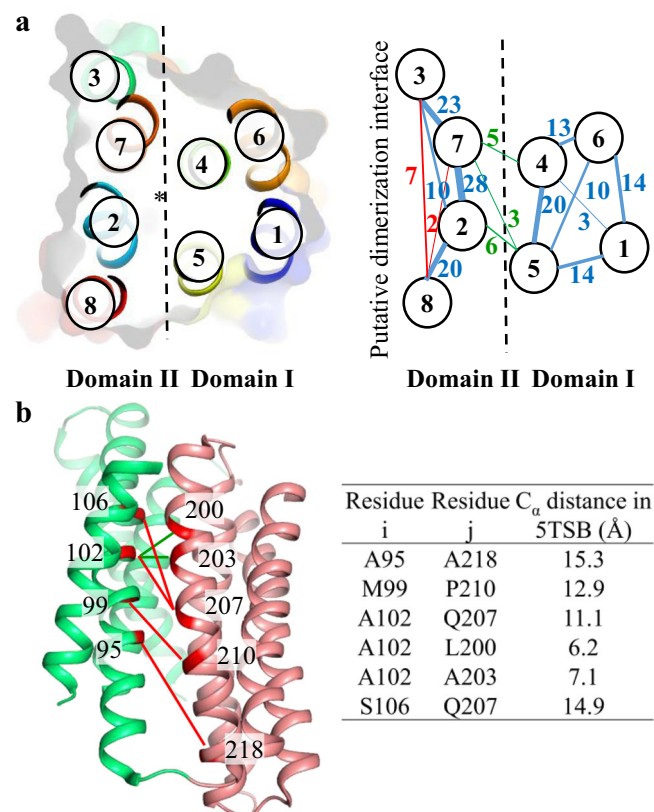

**Fig. 4 | Evolutionary covariance analysis of BbZIP. a** Mapping of the predicted interhelical interactions (with probability >75%) by Evcouplings to the structure of BbZIP. *Left*: The cross-section view of BbZIP in cartoon and surface modes are labeled with TM numbers. The structure is divided by the dashed line into two domains with the asterisk marking the transport pathway. *Right*: The interhelical interactions are indicated by the lines connecting the involved TMs. The numbers of interhelical interactions are labeled beside the lines, and the thickness of the lines is proportional to the number of interhelical interactions. The intradomain interactions within Domain I or Domain II are in blue, the interdomain interactions are in green, and the predicted interactions allowed only in dimeric BbZIP are in red. **b** Mapping of the interactions between α2 and α5 to the IFC structure of BbZIP (PDB entry ID: 5TSB). The interactions consistent with the IFC structure are shown as green lines, whereas the interactions inconsistent with the IFC structure are shown as red lines. The distances of the $C_\alpha$ atoms of the involved residues in 5TSB are listed in the table.

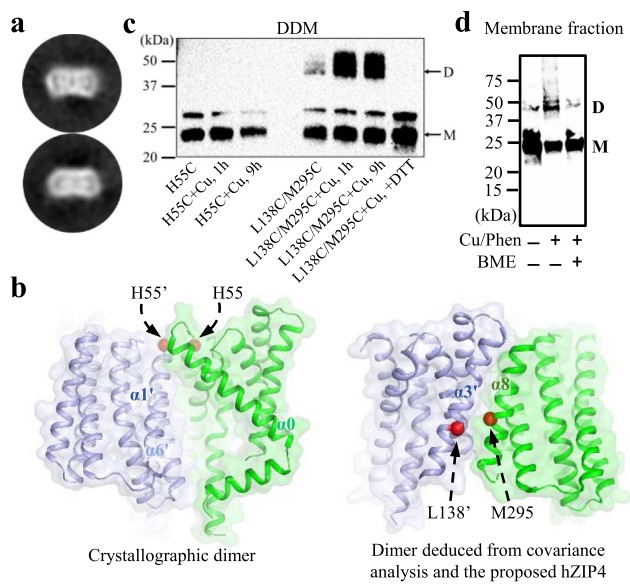

**Fig. 5 | Dimerization of BbZIP. a** Cryo-EM of BbZIP in Amphipol A8-35. Near symmetrical dimer can be visualized in selected 2D images. **b** Comparison of the crystallographic dimer observed in the apo state structure and the dimer deduced from the covariance analysis and the previous computational study on human ZIP4. BbZIP is shown in cartoon and surface modes. The $C_\alpha$ atoms of the residues subjected to mutation into cysteine are shown in red spheres. **c** Chemical crosslinking of the purified H55C and L138C/M295C variants using Cu(II)·(1,10-phenanthroline)$_3$ complex (Cu/Phen) under the indicated conditions. Non-reducing SDS-PAGE and Western blot using the anti-Histag antibody were conducted to detect disulfide bond mediated dimerization. **d** Chemical crosslinking of the L138C/M295C variant in the membrane fraction using the Cu/Phen complex. Monomeric (M) and dimeric (D) species are indicated. Source data are provided as a Source Data file.

shown to form a dimer at pH 4.0[43]. However, BbZIP was crystallized in lipidic cubic phase as a monomer at pH 7.5[46] and it was speculated that the protein was a mixture of monomer and dimer under this condition. To clarify the dimerization issue of BbZIP, we reconstituted the DDM-solubilized BbZIP into Amphipol A8-35 and used single-particle cryo-EM to further investigate dimerization. Albeit of low resolution due to the small particle size and the ambiguity in particle orientation, the selected subset of 2D classification images revealed two near symmetrical objects embedded in the BbZIP-Amphipol complex, indicative of a homodimer (Fig. 5a). When BbZIP in the apo state was crystallized at pH 4.0, two protein molecules in one asymmetric unit were found to associate with each other through α0, α1, and α6 (Fig. 5b), which is, however, on the opposite side of the dimerization interface deduced from the covariance analysis (Fig. 4). To examine which dimerization mode most likely represents the oligomeric state of BbZIP at neutral pH, we generated and purified two variants, H55C and L138C/M295C, for chemical crosslinking experiments. As shown in Fig. 5c, the cysteine residue introduced at the crystallographic dimerization interface (H55C) did not yield dimer upon oxidation by Cu(II)·(1,10-phenanthroline)$_3$ complex, whereas the L138C/M295C

variant, where the two cysteine residues were introduced in α3 and α8 according to the covariance analysis and the human ZIP4 dimerization model, was readily crosslinked via disulfide bond to form a dimer under the same condition. To further validate the dimerization interface, the crosslinking experiment was also conducted in the membrane fraction of the *E.coli* cells expressing the L138C/M295C variant. Similar to the result performed on the purified proteins, a dimerization band can be clearly identified after, but not before, the reaction (Fig. 5d). These results confirm that Domain II (α2/3/7/8) mediates BbZIP dimerization, whereas the crystallographic dimer is likely to be an artifact caused by crystal packing and/or low pH. This dimerization interface is also consistent with a BbZIP dimer structure model predicted by AlphaFold (Supplementary Fig. 3).

**Generation of the outward-facing conformation (OFC) model**

Inverted repeats in the transmembrane domain have been found in many membrane transport proteins[56–58]. It was shown that swapping the residues between the symmetry-related repeats (i.e. repeat-swap homology modeling) allowed the generation of the structural models in alternative conformations[59]. The BbZIP structure consists of two pairs of inverted repeats: α1-α3 & α6-α8, and α4 & α5 (Fig. 6a)[46]. Amino acid sequence comparison showed that the corresponding TMs share considerable sequence identity (Fig. 6b). By following the protocol of repeat-swap homology modeling and then a 100-ns MD simulation, we generated an alternative conformation of BbZIP. Structural inspection showed that the transport site in the model is exposed to the extracellular side and the previously widely opened metal release pathway to the cytoplasm has been drastically narrowed by hydrophobic residues (L92, M99, F170, V272, and P279), A214, A218, and R166, indicating that the model represents an OFC (Fig. 6c–e). S106 and Q207, which are predicted to contact according to the covariance analysis

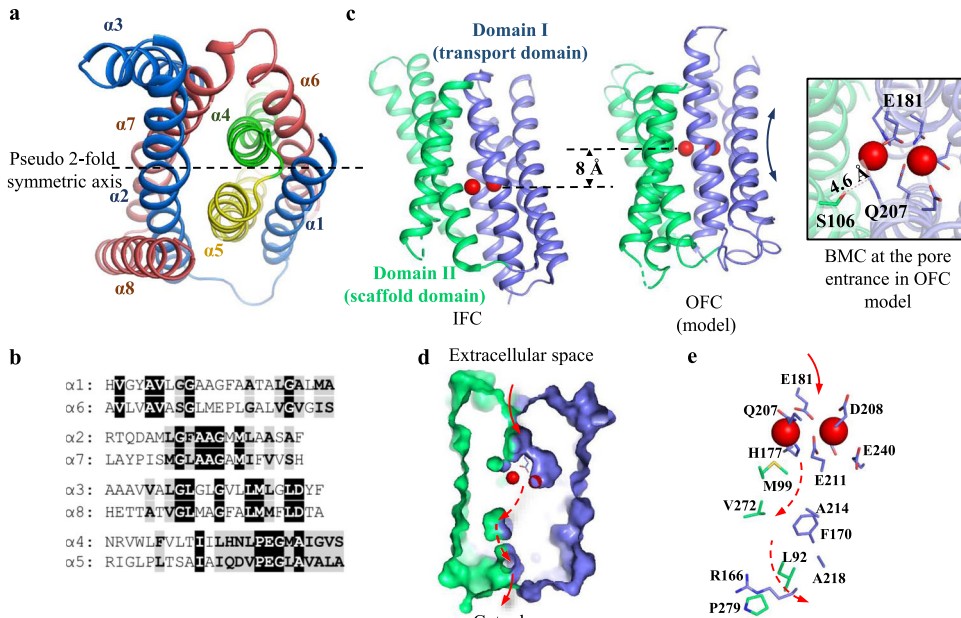

**Fig. 6 | The OFC structural model generated by repeat-swap homology modeling. a** Two pairs of inverted repeats in the structure of BbZIP. α1-α3 (blue) and α6-α8 (dark pink) are symmetrically related by the two-fold pseudo symmetric axis (dashed line). α4 (green) and α5 (yellow) are also related by the same axis. **b** Sequence alignment of the symmetrically related TMs. **c** Comparison of the OFC model and the IFC of BbZIP (PDB entry ID: 5TSA). The metals are depicted as red spheress. Structures are aligned via Domain II (green). The BMC moves up by approximately 8 Å when BbZIP switches from IFC to OFC. The inset shows the top view of the entrance of the transport pathway where S106 approaches Q207 to form a putative metal binding site. **d** Cross-section view of the OFC model in surface mode. The transport pathway is indicated by red arrows: the solid arrow means that the pathway is open to the exterior of the protein, whereas the dashed arrows indicate that the pathway is too narrow to allow metals to pass through. D181, which is the one closest to the extracellular space among the metal chelating residues in the BMC, is shown in stick mode. **e** The zoomed-in transport pathway in the OFC. The residues in the transport site and along the metal release pathway are labeled and shown in stick mode.

but separated by 14.9 Å in the IFC (Fig. 4b), are now close in space and possibly forming a metal binding site at the entrance of the transport pathway (Fig. 6c). Remarkably, structural comparison between the IFC and the OFC model revealed the same two-domain architecture as suggested by the apo state structure (Fig. 3) and the covariant analysis (Fig. 4). When the transporter switches from IFC to OFC, Domain I slides upward against Domain II with little structural changes in either domain. As nearly all the metal chelating residues in the BMC are within Domain I (except for M99 in α2), the upward movement of Domain I lifts the transport site by approximately 8 Å, not only exposing the high affinity metal binding sites to the other side of the membrane but also shortening the distance for metal substrate traveling from the extracellular space to the transport site (Fig. 6c). M99 in α2 may play a role in stabilizing the IFC, and this function in the OFC may be replaced by S106 in α2 which appears to participate in metal chelation at the extracellular side. Nevertheless, the rigid body movement of Domain I relative to Domain II and the vertical translocation of the transport site during the IFC-OFC interconversion are reminiscent of the characteristic conformational changes of elevator-type transporters[60,61].

### Experimental validation of the OFC model

Next, we experimentally examined the computationally generated OFC model to exclude potential artifacts that might be generated during modeling.

First, cysteine accessibility assays were conducted at eight selected positions along the transport pathway. Among the eight residues subjected to mutagenesis to cysteine, three (A184, L200, and A203) are located at the extracellular side of the transport pathway but buried in the IFC; Two (L92 and A214) are exposed to solvent in the IFC; and three (A102, Q207, and V272) are in the middle of the transport pathway. As BbZIP has no endogenous cysteine residue, the

introduced single cysteine in each variant provides the only thiol group to react with the N-ethylmaleimide (NEM), a membrane permeable thiol reacting reagent which has been used to study solvent accessibility of cysteine residues in membrane proteins[62,63]. The membrane fraction containing the expressed cysteine variants was incubated with NEM, and then with monofunctional PEG maleimide 5k (mPEG5K) under denaturing condition. By comparing the relative amount of NEM-modified and mPEG5K-modified species in Western blot, the cysteine accessibility to solvent can be estimated. The results summarized in Fig. 7a showed that, except for those in the middle of the transport pathway, the other tested variants were readily modified by NEM under the native condition, indicating that these residues, including A184, L200, and A203 which are buried in the IFC, can be similarly exposed to a hydrophilic open space to allow NEM to enter and react with the introduced cysteine residues. This result strongly indicates that, besides the experimentally determined IFC, BbZIP must adopt alternative conformation(s) so that the otherwise buried residues can be labeled with NEM. In the OFC model, these three residues are indeed exposed to the extracellular space (Fig. 7a). When the long and highly hydrophilic mPEG5K molecules were directly applied to the purified variants (L92C, L200C, A203C, and A214C), cysteine PEGylation occurred for all the tested variants at comparable levels (Supplementary Fig. 4a), confirming that these residues are indeed similarly exposed to an aqueous environment. To examine whether substrate binding affects the equilibrium between the IFC and the alternative conformation(s), we conducted cysteine accessibility assay on two variants (L200C and A203C) in the absence and presence of added 50 μM ZnCl$_2$. As indicated by the increased percentage of mPEG5K-modified species (Fig. 7b), zinc ions reduced NEM modification at the transition point during the NEM titration. This result suggests that substrate binding favors the conformation(s) (such as the IFC) where the cysteine residues in the variants of L200C and A203C are less

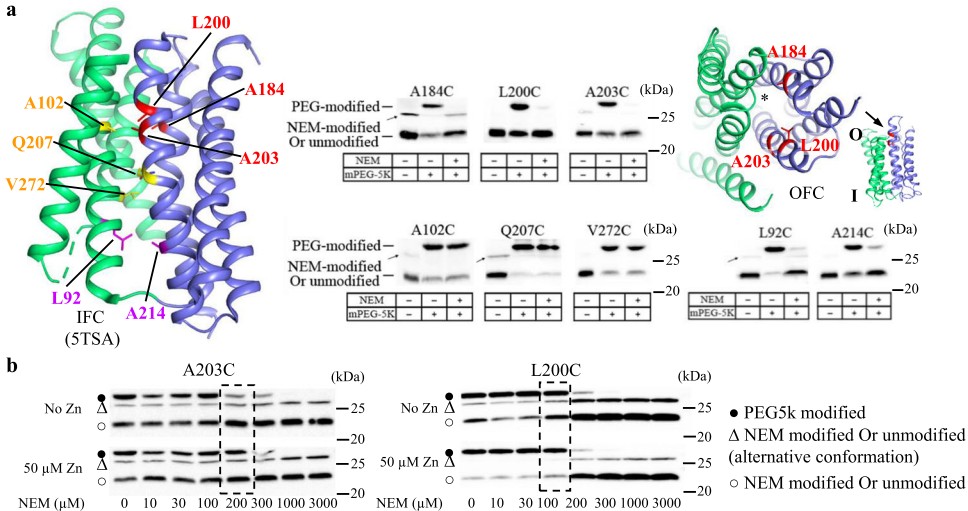

**Fig. 7 | Cysteine accessibility assay of BbZIP in the native membrane. a** Cysteine accessibility assay of BbZIP single cysteine variants. The residues subjected to mutagenesis to cysteine are mapped on the IFC structure (PDB entry ID: 5TSA). The membrane fraction of the cells expressing single cysteine variants was treated with or without NEM under native conditions, and then treated with mPEG5K under denaturing conditions. The samples were applied to SDS-PAGE and Western blot to detect mPEG5K-labeled (PEG-modified) and unlabeled (or NEM labeled) BbZIP. A significant reduction of mPEG5K labeling after NEM treatment indicates an access of the cysteine to NEM under native condition. The residues buried in the IFC but accessible to NEM are shown in red. The residues exposed to the solvent in the IFC structure and accessible to NEM are in magenta. Those inaccessible to NEM are in yellow. A184, L200, and A203 are also mapped on the OFC model in a tilted top view along the direction indicted by the arrow in the side view of the OFC model (inset). The asterisk indicates the transport pathway. **b** Effects of metal substrate on cysteine accessibility of selected BbZIP variants. The procedure is the same as in (**a**) except that the membrane fraction was treated with NEM at indicated concentrations in the absence or presence of added 50 μM ZnCl₂. The dashed boxes highlight the conditions under which the efficiency of NEM labeling was reduced (as reflected by the relatively increased mPEG5K labeling) by zinc ions. Source data are provided as a Source Data file.

accessible to NEM. The modest effect may imply that the transporter undergoes a constant conformational switch and the equilibrium among the conformers can be moderately modulated by substrate binding/release.

Next, mercury (Hg)-mediated chemical crosslinking was conducted to validate the OFC model. It has been reported that crosslinking of two cysteine residues by Hg²⁺ may alter the migration of the target membrane protein in SDS-PAGE[64,65]. We introduced two cysteine residues at A95 in α2 (Domain II) and L217 in α5 (Domain I), of which the Cβ atoms are 13.3 Å apart in the IFC but only 4.9 Å in the OFC model (Fig. 8a). Both residues are located at the interface of Domain I and Domain II. To form a S-Hg-S linkage between two cysteine residues, the maximum allowed distance is approximately 7.8 Å. The A95C/L217C variant was purified, treated with HgCl₂, and applied to non-reducing SDS-PAGE. As shown in Fig. 8a, the migration of the A95C/A217C variant was clearly changed after the HgCl₂ treatment and the band shift was reversed by adding reducing agent 2-mercaptoethanol after crosslinking. The shift was also prohibited by pre-treatment with the thiol reacting agent NEM. The same band shift after HgCl₂ treatment was also observed for the A95C/A217C variant, but not for the wild type protein, in the native membrane (Supplementary Fig. 4b). Removal of residual Hg²⁺ in sample solution via size-exclusion chromatography did not affect band shift (Supplementary Fig. 4c), excluding the possibility that the crosslinked species formed merely in the SDS sample loading buffer under a denaturing condition. In contrast, none of the single cysteine variants (A95C or L217C) exhibited band shift with the same treatment. These results indicated that the band shift of the double variant was exclusively attributed to Hg²⁺ reacting with both cysteine residues to form a S-Hg-S linkage. Using the same approach, we examined additional three double cysteine variants with cysteine substitutions occurring in α4 and α7. As shown in Fig. 8b, the treatment with HgCl₂ led to a band shift for two variants (V167C/V272C and L169C/V272C), but not for the variant of W168C/V272C. Although all three residues (167, 168, and 169 in α4 at Domain I) are predicted to approach residue 272 (in α7 at Domain II) in the OFC (but not in the

IFC), residue 168 is exclusively located at the side of α4 which faces away from residue 272 according to the OFC model, which likely explains the lack of crosslinking for the W168C/V272C variant. Collectively, chemical crosslinking experiments indicate that Domain I must significantly move upward relative to Domain II as predicted by the OFC model so that the two otherwise distal residues approach to each other to allow the formation of the Hg-mediated crosslinking, indicative of an elevator-like movement.

## Crucial roles of small residues at the domain interface

The hallmark of elevator-type transporters is the rigid body sliding of the transport domain, which exclusively (or almost exclusively) carries substrate(s), against the static scaffold domain. This movement is facilitated by a smooth and generally hydrophobic interface between the two domains. For example, a "greasy" domain interface consisting primarily of small residues was observed in CitS, a Na⁺/citrate symporter and an established elevator transporter[66]. Similarly, structural inspection revealed that, out of approximately forty residues at the interface between the two domains of BbZIP, nearly half of them are small residues (Ala, Gly, or Ser) (Supplementary Fig. 5). Multiple sequence alignment of ZIPs from different subfamilies identified four highly conserved small residues (A95, A184, A203, and A214 in BbZIP, Fig. 9a, b) at the domain interface. To examine the importance of these residues for zinc transport, we individually substituted each of the corresponding residues with a valine (hydrophobic with a slightly bigger sidechain) and a phenylalanine (hydrophobic with a bulky and rigid sidechain) in human ZIP4 (A386, A514, A532, and G543), a well-characterized ZIP for which the cell-based zinc transport assay has been frequently conducted in recent studies[34,48,55,67–69]. As shown in Fig. 9c, substitution of A386 and A532 with valine greatly reduced the zinc transport activity by more than 80% and substitution of A532 with phenylalanine completely eliminated transport activity. Substitution of A514 and G543 with valine also significantly reduced activity but to a lesser extent. Phenylalanine substitution of A514 led to a greater activity suppression, but the G543F variant unexpectedly exhibited an

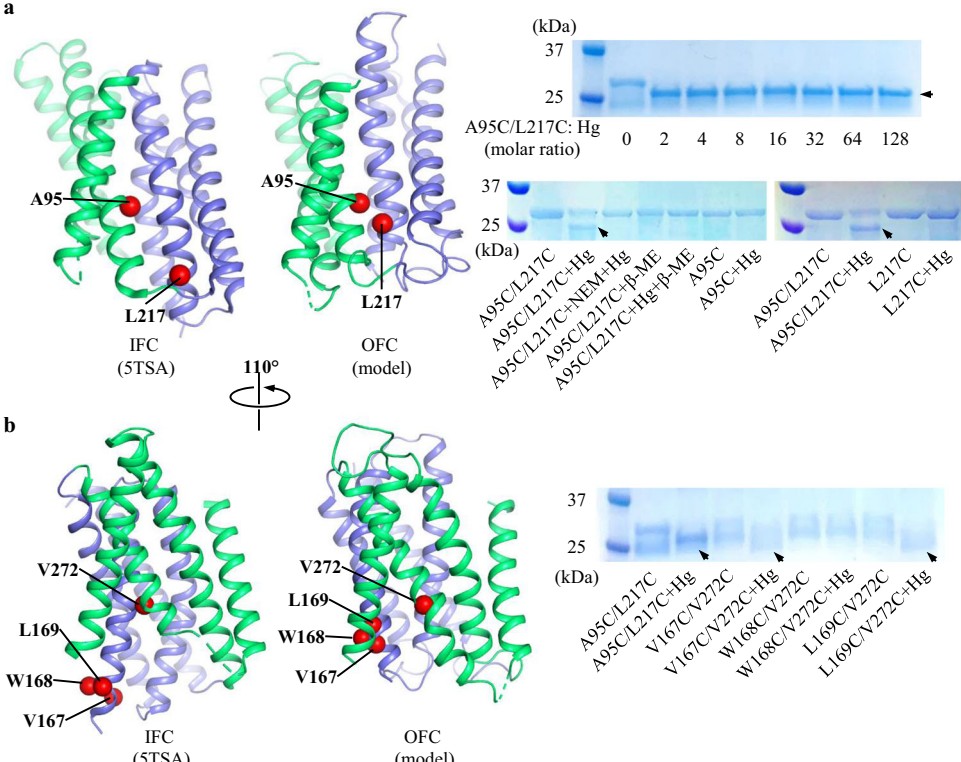

**Fig. 8 | Hg$^{2+}$-mediated chemical crosslinking of BbZIP to validate the proposed OFC. a** Crosslinking of the A95C/L217C variant. The protein was purified in DDM, incubated with HgCl$_2$ and then applied to non-reducing SDS-PAGE. The shifted bands after Hg treatment are indicated by arrows. The same batch of the purified protein was treated differently as indicated to confirm that the band shift is attributed to cysteine reacting with Hg$^{2+}$. The single cysteine variants (A95C and L217C) were treated in the same way as the double variant. **b** Crosslinking of the double cysteine variants (V167C/V272C, W168C/V272C, and L169C/V272C) with mutations on the opposite side of residues 95 and 217. Note that the W168C/V272C variant could not be crosslinked by HgCl$_2$ under the same condition. The residues subjected to mutagenesis are mapped to the IFC and the OFC model, respectively. The C$_\beta$ atoms of the residues are depicted in sphere mode. Source data are provided as a Source Data file.

activity similar to the wild-type ZIP4. Compared with A386 and A532, A514 and G543 are at the peripheral region of the domain interface (Fig. 9b), which may explain why substitution with bulky amino acids at these two positions caused less disruption of the transporter's function. The same pattern was also observed when these residues were replaced with cysteine residues (Supplementary Fig. 8). Overall, these results demonstrated the importance of the conserved small residues at the domain interface and supported the notion that a smooth domain interface is required for optimal activity of an elevator transporter, which is in line with the previous reports that mutations at the domain interface of elevator transporters may drastically influence transport activity[70–72].

## Discussion

The ubiquitously expressed ZIP family members play vital roles in *d*-block metal homeostasis, and are also linked with cancers, osteoarthritis[27], hypertension[29], B cell related diseases[30], and fungal infections[73,74]. Clarifying the transport mechanism will not only promote the understanding of metal homeostasis but also facilitate drug discovery and engineering of ZIPs for potential applications in agriculture and environmental protection. In this work, we solved the structure of BbZIP in the apo state, filling a missing piece in the transport cycle. More importantly, by combining the efforts of structural biology, evolutionary covariance analysis, computational modeling, biochemical assays, and functional studies, we present compelling evidence supporting the elevator-type transport mode for BbZIP, a representative prokaryotic ZIP.

Several lines of evidence collectively support an elevator-type transport mode for BbZIP. (1) The apo state structure revealed a rigid body movement of Domain I (α1/4/5/6) relative to Domain II (α2/3/7/8) (Fig. 3). (2) This two-domain architecture also emerged in the evolutionary covariance analysis (Fig. 4). Some of the predicted interactions between the two domains are not consistent with the IFC, implying the presence of alternative conformation(s). (3) The OFC model generated by repeat-swap homology modeling, which was experimentally validated by cysteine accessibility assay and chemical crosslinking in both detergents and the native membranes (Figs. 7, 8), unraveled a significant upward rigid body movement of Domain I relative to Domain II when the transporter switches from the IFC to the OFC model (Fig. 6). This movement of Domain I leads to a vertical translocation of the transport site by approximately 8 Å, which is a strong characteristic of the elevator-type transporters. (4) Chemical crosslinking experiment revealed a BbZIP dimer mediated by Domain II (Fig. 5). All these findings are consistent with the general characteristics of an elevator-type transporter for which the transport domain (Domain I in BbZIP) carries the substrate and moves vertically as a rigid body against the static scaffold domain (Domain II) which often mediates oligomerization[61]. As demonstrated in the study of divalent-anion sodium symporters[75], substrate binding neutralizes the net charges of the transport site to facilitate sliding of the transport domain against the scaffold domain via the charge-compensation mechanism which was also proposed for other elevator transporters. Possibly, as the BMC of BbZIP is negatively charged, binding of divalent metal ion substrate may facilitate a conformational switch from the OFC in the apo state to the IFC in the metal-bound state. As metal removal from the transport site of BbZIP leads to disassemble of the primary transport site (M1) and then the swing of α4 away from α3 and α7 (Figs. 2, 3), we postulate that the thus reduced interdomain interactions may be crucial for the IFC-to-OFC

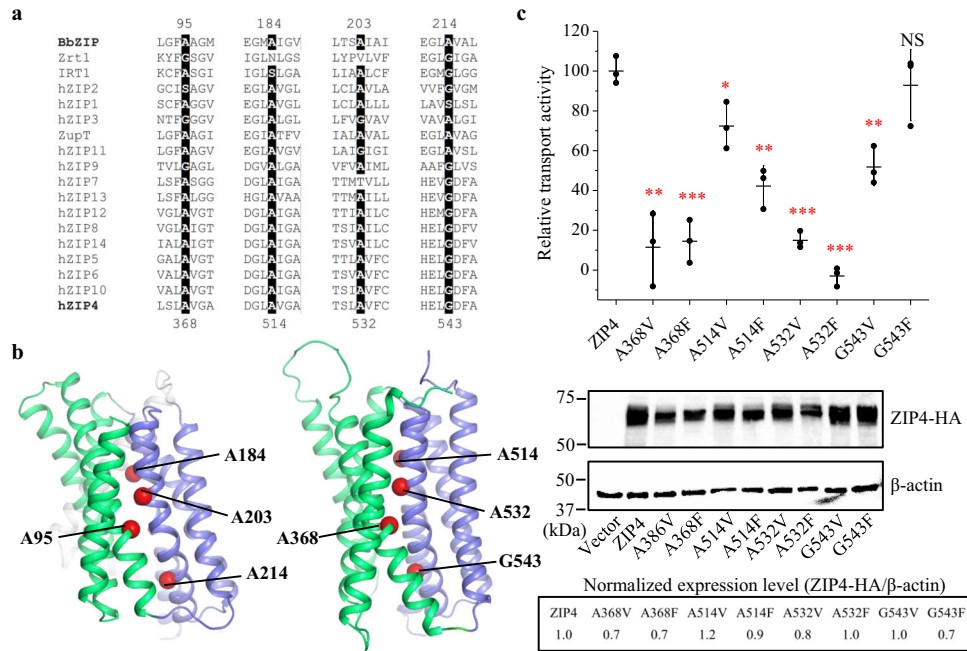

**Fig. 9 | Functional characterization of the conserved small residues at the domain interface. a** Multiple sequence alignment of representative ZIPs from major subfamilies. Conserved small residues (Ala, Gly, or Ser) identified at the interface between the transport domain and the scaffold domain are highlighted. **b** Mapping of the conserved small residues in BbZIP and in human ZIP4. The $C_\alpha$ atoms of the residues chosen for mutagenesis are depicted as red spheres. Only the structure of ZIP4 transmembrane domain (predicted by AlphaFold, https://alphafold.ebi.ac.uk/entry/Q6P5W5) is shown. **c** Zinc transport activities of ZIP4 and the variants. HEK293T cells transiently transfected with ZIP4 or the variants were applied to the radioactive $^{65}$Zn transport assay. The relative transport activity of each variant was calibrated with the normalized expression level, which was estimated in Western blot experiments, and expressed as the percentage of the activity of the wild type ZIP4. The shown data are from one representative experiment and 2-3 independent experiments with similar results were conducted for each variant. Three biological replicates were included for each variant in one experiment. The horizontal bar of the scatter dot plot represents the mean and the vertical bar indicates the standard deviation. The asterisks indicate the significant differences between the variants and the wild type ZIP4 (two-sided Student's $t$ tests: $*P \le 0.05$; $**P \le 0.01$; $***P \le 0.001$; NS—no significant difference). The exact P values are 0.0015, 0.00032, 0.024, 0.0012, 0.000053, 0.000028, 0.002, and 0.56 for the variants of A368V, A368F, A514V, A514F, A532V, A532F, G543V, and G543F, respectively. Source data are provided as a Source Data file.

transition for BbZIP. Consistently, metal-dependent NEM modification of cysteine residues was observed (Fig. 7b). Based on these findings, we propose a transport mechanism of BbZIP which is illustrated in Fig. 10. Of great interest, the structures of fourteen human ZIPs predicted by AlphaFold (Supplementary Fig. 6) cover the whole range of the conformational states revealed by the IFC structures and the OFC model of BbZIP. Whether these distinct conformations represent the structural ensemble of the ZIPs so that the elevator-type transport mechanism is shared by other ZIPs is warranted for further investigation, and the mutagenesis and functional study on human ZIP4 (Fig. 9) already supports this possibility.

Although oligomerization is very common in elevator transporters, exceptions exist, including the bile acid sodium symporter ASBT with a similar wall-like scaffold domain composed of four TMs as observed in BbZIP[76,77]. BbZIP indeed forms a dimer in both detergents and in the native membrane (Fig. 5), but the dimeric form seems to be unstable at least in detergents and in lipidic cubic phase as it has been frequently crystallized in the monomeric form. We have previously shown that the purified protein may be in a rapid monomer-dimer equilibrium in detergents at neutral pH[46]. Oligomerization of the scaffold domain is believed to be beneficial for an elevator transporter's function, whereas a reversible and potentially tunable oligomerization may allow for regulation. For instance, heterodimerization of ZIP6 and ZIP10 has been shown to be important for their functions to promote cell growth[23]. Nevertheless, the transmembrane domain may only partially contribute to dimerization and the extracellular domain (ECD) of some ZIPs, such as the ECD of ZIP4, may play a key role in promoting dimerization for optimal zinc transport[55].

According to a recent classification of the elevator-type transporters[61], BbZIP appears to belong to the moving barrier elevator with two gates. Although a hydrophobic gate within the pathway toward the extracellular space has been noticed in the IFC structures[46,48], whether a functional gate is formed within the metal release pathway when BbZIP is in the OFC is still unclear. Our results strongly indicate that the transporter can adopt an OFC so that the metal ion substrate in the extracellular space can reach the transport site, but whether the pathway toward the cytoplasm is blocked in the OFC needs to be experimentally examined, although the OFC model presented in this work seems to support it. One putative candidate for the gate at the cytoplasmic side consists of R166 in α4 of the transport domain and two residues (H275 and E276) in α7 of the scaffold domain (Supplementary Fig. 7). In the IFC, R166 is distant from H275 and E276, both of which coordinate metal substrates in the release pathway; in the OFC model, R166 approaches the two metal chelating residues to form hydrogen bonds which may stabilize the OFC and contribute to block the transport pathway to prevent leak. If this putative gate (or other residues) blocks the conduit in the OFC, BbZIP would be an authentic carrier protein; if the gate was leaky so that the ion could still pass through when the transport site is open to the extracellular space, it would explain why BbZIP behaves as an ion channel[7,43]. One known example is cystic fibrosis transmembrane conductance regulator (CFTR), which is the only ion channel in the entire ABC transporter family. The leaky internal (cytoplasmic) gate of CFTR exclusively accounts for the chloride ion channel activity[78,79].

Taken together, the evidence presented in this work consistently support an elevator-type transport mode for BbZIP, but we cannot completely exclude the possibility that BbZIP may use an

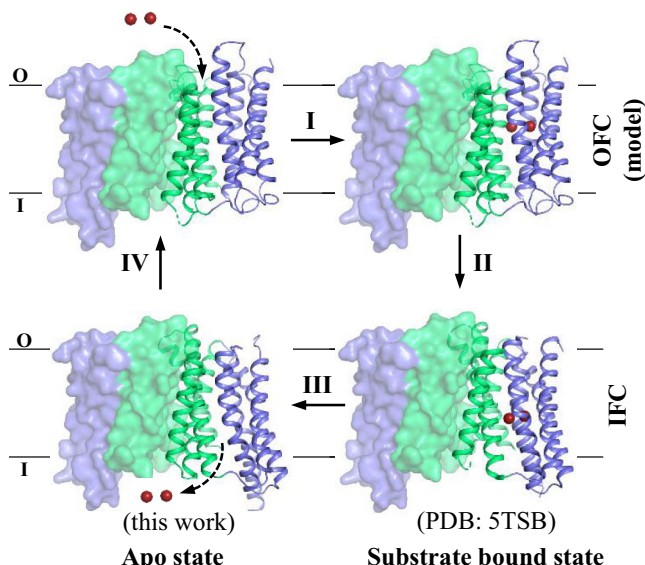

**Fig. 10 | Proposed transport cycle of BbZIP.** The complete transport cycle consists of at least four consecutive steps: I—Metal substrate(s) bind to the transport site(s) of the transporter in the OFC; II—Metal substrate binding facilitates the conformational change from the OFC to the IFC, presumably via a charge-compensation mechanism; III—Metal release from the transport site(s) causes disassemble of the primary transport site (M1); IV - The altered domain interface triggered by M1 disassemble facilitates a conformational switch from the IFC to the OFC. For clarity, the protomer shown in surface mode is fixed in the IFC, and the other protomer in carton mode is shown in distinct conformational states with the scaffold domain (Domain II) colored in green and the transport domains (Domain I) in blue. The metal ion substrates are depicted as dark red spheres. The dashed arrows indicate the directions of metal transport. During transport, the dimeric scaffold domain largely remains static, whereas the transport domain vertically slides against the scaffold domain. The OFC models were generated using repeat-swap homology modeling and validated by chemical crosslinking and cysteine accessibility assay in this work. The IFC structure with bound metal substrate is derived from the previously solved structure (PDB entry ID: 5TSB) and the IFC structure in the apo state is reported in this work.

unprecedented transport mechanism in the absence of an experimentally determined structure representing the OFC state. For comparison, the secondary divalent metal transporters in the ZnT family and the Nramp family use the LeuT-like rocking bundle mechanism[80,81], whereas heavy metal transport by the P-type ATPases are exclusively driven by ATP hydrolysis. Although the nickel/cobalt transporting energy-coupling factor transporters are recently considered to be elevator transporters given the proposed toppling mechanism[61], they are essentially non-canonical ABC transporters composed of three to four subunits and are therefore fundamentally different from the ZIPs in every respect. Accordingly, the elevator-type mechanism utilized by BbZIP and potentially other ZIPs (Supplementary Fig. 6) appears to be unique among the known metal transporters. How this transport mechanism is associated with the biochemical characteristics and the biological functions of the ZIPs is to be investigated in future study.

While this manuscript was under revision, an independent study reported a structure of BbZIP in the metal-free state similarly crystallized at low pH (PDB ID: 7Z6N, referred to as 7Z6N hereafter)[82]. Contrast and comparison of two structures provide additional insights into the elevator-type transport mechanism (Supplementary Fig. 9). Firstly, the two structures solved in different space groups consistently reveal a rigid body rotation of the transport domain against the scaffold domain but with a different relative orientation between them (Supplementary Fig. 9a), providing an additional piece of evidence supporting the two-domain architecture and also excluding the possibility that the rigid body movement is derived from crystallographic

artifacts. Secondly, as revealed by the electron density maps, the cytoplasmic side of α4 appears to be more dynamic in 7Z6N than that in the structure solved in this work (Supplementary Fig. 9b). The well-resolved densities of our structure unambiguously unraveled the conformational rearrangements upon metal release, including disassemble of the primary transport site (M1) and dissociation of the cytoplasmic side of α4 from the scaffold domain (Figs. 2, 3), which are proposed to be critically involved in the conformational switch from the metal-bound IFC to the metal-free OFC (Fig. 10). Thirdly, the observed loose crystallographic dimer in 7Z6N is not present in our crystal structure (Supplementary Fig. 9c). This difference reinforces the notion that the BbZIP dimer is unstable and the purified protein is likely in a monomer-dimer equilibrium in detergents as we proposed previously[46]. Of great interest, our proposed dimer based on covariance analysis and validated by chemical crosslinking in detergents and in the native membrane is largely consistent with the proposed dimer model based on crystal packing analysis and molecular docking in the other report. Lastly, the amphipathic helix (α0a, residues 7–21) solved in our structure was missed in 7Z6N. α0a associates with α4 via hydrophobic interactions and α3 via hydrogen bonds (Supplementary Fig. 9d), which provides a reasonable structural explanation of hyperactivation of the transporter upon α0a deletion which would otherwise limit the movement of the transport domain relative to the scaffold domain[82]. Remarkably, although the method of generating the OFC model in this work (repeat-swap homology modeling) is different from the approach used in that report (manually adjusting the position of the transport domain relative to the scaffold domain to eliminate the mismatch between the predicted contacting residues and the structure in the IFC), the two OFC models are consistent in that the transport site is lifted toward the extracellular space by ~8 Å upon the IFC-to-OFC transition (Fig. 6). Critically, our proposed OFC model has been experimentally validated in this work by complementary biochemical approaches conducted both in detergents and in the native membrane (Figs. 7, 8). Overall, the two studies independently and consistently support the elevator-type transport mechanism for BbZIP. The functional studies on human ZIP4 suggest that this mechanism is likely shared by other ZIP family members (Fig. 9).

## Methods

### Genes, plasmids, mutagenesis, and reagents

The gene encoding BbZIP (National Center for Biotechnology Information reference code: WP_010926504) was synthesized with optimized codons for *Escherichia coli* (Integrated DNA Technologies) and inserted into the pLW01 vector with a thrombin cleavage site inserted between the N-terminal His-tag and BbZIP[46]. The complementary DNA of human *zip4* (GenBank access number: BC062625) from Mammalian Gene Collection were purchased from GE Healthcare. The ZIP4 coding sequence was inserted into a modified pEGFP-N1 vector (Clontech) in which the downstream EGFP gene was deleted and an HA tag was added at the C-terminus. Site-directed mutagenesis was conducted using the QuikChange Mutagenesis kit (Agilent). Primers used for mutagenesis are listed in Supplementary Table 3.1-Oleoyl-rac-glycerol (monoolein), N-ethylmaleimide (NEM), 1,10-phenanthroline, Tris(2-carboxyethyl)phosphine (TCEP) were purchased from Sigma-Aldrich. Thrombin was purchased from Novagen. Monofunctional PEG maleimide 5k (mPEG5K) was purchased from Creative PEGWorks.

### Protein expression and purification

The expression of BbZIP was reported previously[46]. In brief, the wild type or the variants of BbZIP was expressed in the strain of C41 (DE3) pLysS (Lucigen) in LBE-5052 Autoinduction medium for 24 h at room temperature. After harvest, spheroplasts were prepared and lysed in the buffer containing 20 mM Hepes (pH 7.3), 300 mM NaCl, 0.25 mM $CdCl_2$, and cOmplete protease inhibitors (Sigma-Aldrich)[83]. n-Dodecyl-β-D-maltoside (DDM, Anatrace) was added to

solubilize the membrane fraction at the final concentration of 1.5% (w/v). The His-tagged protein was purified using HisPur Cobalt Resin (Thermo Fisher Scientific) in 20 mM Hepes (pH 7.3), 300 mM NaCl, 5% glycerol, 0.25 mM CdCl$_2$, and 0.1% DDM. The sample was then concentrated and loaded onto a Superdex Increase 200 column (GE Healthcare) equilibrated with the gel filtration buffer containing 10 mM Hepes, pH 7.3, 300 mM NaCl, 5% glycerol, 0.25 mM CdCl$_2$, and 0.05% DDM. For the variants with introduced cysteine residue(s), 1 mM TCEP was added during purification but excluded in the gel filtration buffer. The peak fractions were used for crystallization, cryo-EM, or crosslinking experiments.

### Crystallization, data collection, and structure determination
Purified BbZIP was concentrated to 15 mg/ml and then mixed with the molten monoolein with two coupled syringes at a ratio of 2:3 (protein/ monoolein, v/v). All crystallization trials were set up using a Gryphon crystallization robot (Art Robbins Instruments). 50 nl of BbZIP-monoolein mixture covered with 800 nl of well solution was sandwiched with lipidic cubic phase sandwich set (Hampton Research). Stick-shaped crystals appeared after one week under the condition containing 27% PEG200, 20 mM NaAc, pH 4.0, 0.2 M (NH$_4$)$_2$SO$_4$, 20 mM NaCl at 21 °C and grew to full size in two weeks. Crystals were harvested with a MiTeGen micromesh and flash-frozen in liquid nitrogen.

The X-ray diffraction data were collected at the General Medicine and Cancer Institutes Collaborative Access Team (GM/CA-CAT) (23-ID-B/D) at Advanced Photon Source (APS). The diffraction datasets were indexed, integrated, and scaled in HKL2000 v712. The apo state structure was solved in molecular replacement using the previously solved structure (PDB: 5TSB) as the search model in Phenix v1.10.1. Iterative model building and refinement were conducted in COOT v0.8.2 and Phenix, respectively. All figures of protein structures were generated by PyMOL v1.3 (Schrödinger LLC).

### Single particle Cryo-EM
Purified BbZIP was mixed with Amphipol A8-35 (Anatrace) at a mass ratio of 1:3 (BbZIP:Amphipol) and allowed to incubate with gentle shaking at 4 °C overnight. Damp Bio-Beads SM-2 resin (Bio-Rad) was added to the mixture for 3 h at 4 °C, after which the supernatant was collected and filtered through a 0.2 mm spin filter. The sample was then loaded onto a Superdex Increase 200 column (GE Healthcare) equilibrated with the gel filtration buffer (20 mM Hepes, pH 7.3, 200 mM NaCl). The peak fractions were pooled and concentrated to 2 mg/ml. Sample (3 µl) was loaded onto a glow-discharged QUAN-TIFOIL grids (Q250-CR1.3, EMS) and flash-frozen using Vitrobot (Mark IV, Thermo Fisher Scientific). Cryo-EM data for BbZIP in Amphipol A8-35 were collected at University of Michigan, Life Sciences Institute using a 300 kV Krios Titan (FEI/ThermoFisher) electron microscope equipped with a Bioquantum (Gatan), SerialEM[84], a nominal magnification of 81,000x, a calibrated pixel size of 1.1 Å/pixel, and a dose rate of 15 electrons/Å2/s. Movies were recorded at 100 ms/frame for 3 s (30 frames), resulting in a total radiation dose of 45 electrons/Å2. Defocus range was varied between −1.5 µm and −2.5 µm. A total of 16,134 micrographs were recorded. Dose weighting, motion correction was performed using Warp[85]. Subsequent image processing was performed using cryoS-PARC v3.3.2[86]. Blob picker yielded an initial set of 12,522,072 particles. Particles were binned 2x, extracted into 180 × 180 pixel boxes, and subjected to multiple rounds of reference-free 2D classification and removal of poorly populated classes. A brief flowchart for data processing is shown in Supplementary Fig. 10.

### Repeat-swap homology modeling to generate the OFC model
The established protocol for repeat-swap homology model was followed to generate the OFC model[59]. In brief, the amino acids of the aligned TMs (α1-α3 vs. α6-α8, and α4 vs. α5. See sequence alignment in Fig. 6. The full sequence alignment is shown in Supplementary Fig. 11.) were swapped between the corresponding elements in the IFC structure (PDB entry ID: 5TSA) to generate an initial OFC model using MODELLER v9.20[87]. The segments which are located in TM but could not be aligned, including residues 105-112 in α2 and 274-282 in α7, were modeled as α-helices, whereas the loops connecting the TMs were built de novo. The resulting structure model was subjected to 100 ns MD simulation in AMBER (v18, force field ff19SB) for energy minimization[88].

### Structure prediction by AlphaFold
Structures predicted by AlphaFold were retrieved from the AlphaFold Protein Structure Database [https://alphafold.ebi.ac.uk/] for human ZIPs or generated by AlphaFold Colab [https://colab.research.google.com/github/deepmind/alphafold/blob/main/notebooks/AlphaFold.ipynb] for BbZIP dimer.

### Cysteine accessibility assay
The spheroplasts-derived membrane fractions of the *E.coli* cells expressing BbZIP variants were incubated with 3 mM NEM for 1 h at 4 °C, and then washed twice to remove excessive NEM with the buffer containing 100 mM Tris, pH 7.0, 60 mM NaCl, 10 mM KCl through centrifugation. The membrane fraction was then collected and dissolved with the denaturing buffer containing 6 M urea, 0.5% SDS, and 0.5 mM DTT (to quench residual NEM) by gentle shaking for 15 min at room temperature, and 5 mM mPEG5K (final concentration) was then added to react with unmodified cysteine residue for 1 h at room temperature. For the samples without NEM treatment, the same membrane fractions were dissolved with the denaturing buffer and treated with 5 mM mPEG5K as described above. All the samples were mixed 4xSDS-PAGE sample loading buffer containing 20% β-mercaptoethonal (β-ME) and subjected to SDS-PAGE. The BbZIP variants were detected in Western blot using an anti-Histag antibody at 1:5000 dilution (Invitrogen, Catalog# 37-2900) and an HRP-conjugated anti-mouse immunoglobulin-G antibody at 1:5000 dilution (Cell Signaling Technology, Catalog#7076 S). The images of the blots were taken using a Bio-Rad ChemiDoc Imaging System.

To test the effects of metal substrate on cysteine accessibility, the selected variants in the native membrane were subjected to NEM titration with indicated concentrations (0–3 mM) in the presence and absence of 50 µM ZnCl$_2$ for 1 h at 4 °C. The other procedures were the same as described above.

Cysteine accessibility assay was also performed using mPEG5K as the thiol reacting reagent. Selected purified proteins at 10−20 µM were treated with 1 mM mPEG5K for 0 or 30 min in the presence of 3 mM EDTA. The reaction was terminated by 100 mM water-soluble thiol reacting reagent methyl methanethiosulfonate before analysis in SDS-PAGE.

### Chemical crosslinking
**Hg-mediated chemical crosslinking.** The BbZIP variants (A95C/ L217C, V167C/V272C, W168C/V272C, and L169C/V272C) were purified in the same way as the wild type protein with 1 mM TCEP included in the buffers throughout the purification except for the last step of size-exclusion chromatography. The purified proteins at approximately 10 µM were incubated with HgCl$_2$ at the indicated molar ratios for 30 min at room temperature. The reactions were terminated by the addition of 2 mM NEM to block the unreacted cysteine residues, and the samples were applied to non-reducing SDS-PAGE. The A95C and L217C single cysteine variants were treated in the same manner. To examine whether the gel shift in SDS-PAGE was caused by the reaction of the cysteine residues with Hg$^{2+}$, the crosslinking product was incubated with 2 mM β-ME for 10 min or the protein sample was pretreated with 2 mM NEM for 10 min prior to HgCl$_2$ treatment.

Hg-mediated crosslinking experiment was also conducted in the native membrane. The spheroplasts-derived membrane fraction of *E.coli* cells expressing the A95C/L217C variant was incubated with 50 μM HgCl₂ at room temperature for 1 h with gentle shaking. The following procedure was the same as described above for purified proteins.

**Disulfide bond crosslinking.** Reactions were performed on purified proteins and also in the native membrane. The variants (H55C and L138C/M295C) were purified in DDM with 1 mM TCEP except for in the last step of size-exclusion chromatography, then incubated with 50 μM Cu(II)-(1,10 phenanthroline)3 complex (final concentration) made freshly by mixing 1 mM CuSO₄ and 3 mM 1,10-phenanthroline. The reaction was carried out at 25 °C for the indicated time and terminated by adding SDS-PAGE sample loading buffer containing 20 mM EDTA. The samples were subjected to non-reducing SDS-PAGE, followed by Western blot using the anti-Histag antibody. To conduct crosslinking experiment in the native membrane, the spheroplasts-derived membrane fraction of the *E.coli* cells expressing the L138C/M295C variant was incubated with 1 mM of Cu(II)-(1,10 phenanthroline)3 complex at 25 °C for 1 h. The membrane fraction was then washed to remove the copper complex before being mixed with the SDS sample loading buffer. The crosslinked products were then detected in Western blot by using a custom monoclonal antibody. This antibody was raised by injecting purified BbZIP into mouse, which was performed by Creative Biolabs Inc. The sensitivity and specificity of this antibody were demonstrated in Supplementary Fig. 12. An HRP-conjugated anti-mouse immunoglobulin-G antibody (Cell Signaling Technology, Catalog# 7076 S) was used as the secondary antibody at 1:5000 dilution in Western blot.

**Mammalian cell culture, transfection, Cell-based zinc transport assay, and Western blot**
Human embryonic kidney cells (HEK293T, ATCC, Catalog# CRL-3216) were cultured in Dulbecco's modified eagle medium (DMEM, Thermo Fisher Scientific) supplemented with 10% (v/v) fetal bovine serum (FBS, Thermo Fisher Scientific) and Antibiotic-Antimycotic solution (Thermo Fisher Scientific) at 5% CO₂ and 37 °C. Cells were seeded on the polystyrene 24-well trays (Alkali Scientific) for 16 h in the basal medium and transfected with 0.8 μg DNA/well using lipofectamine 2000 (Thermo Fisher Scientific) in DMEM with 10% FBS.

The zinc transport activities of ZIP4 and the variants were tested using the cell-based transport assay. Twenty hours post-transfection, cells were washed with the washing buffer (10 mM HEPES, 142 mM NaCl, 5 mM KCl, 10 mM glucose, pH 7.3) followed by incubation with Chelex-treated DMEM media (10% FBS). 5 μM Zn²⁺ (0.05 μCi/well) was added to cells. After incubation at 37 °C for 30 min, the plates were transferred on ice and the ice-cold washing buffer with 1 mM EDTA was added to stop metal uptake. The cells were washed twice and pelleted through centrifugation at 120 × *g* for 5 min before lysis with 0.5% Triton X-100. A Packard Cobra Auto-Gamma counter was used to measure radioactivity. The transport activity was determined by subtracting the radioactivities of ⁶⁵Zn associated with the cells transfected with the empty vector from those associated with the cells transfected with metal transporters. The results are shown in Fig. 9, S8.

For Western blot, the samples mixed with the SDS sample loading buffer were heated at 96 °C for 10 min before loading on SDS-PAGE gel. The protein bands were transferred to PVDF membranes (Millipore). After being blocked with 5% nonfat dry milk, the membranes were incubated with a mouse anti-HA antibody at 1:5000 dilution (Invitrogen, Catalog# 26183) at 4 °C overnight. As loading control, β-actin levels were detected using a rabbit anti-β-actin antibody at 1:5000 dilution (Cell Signaling Technology, Catalog# 4970S). Bound primary

antibodies were detected with an HRP-conjugated anti-mouse immunoglobulin-G at 1:5000 dilution (Cell Signaling Technology, Catalog# 7076S) for ZIP4 or an HRP-conjugated anti-rabbit immunoglobulin-G for β-actin at 1:5000 dilution (Cell Signaling Technology, Catalog# 7074S) by chemiluminescence (VWR). The images of the blots were taken using a Bio-Rad ChemiDoc Imaging System.

### Statistics and reproducibility
Statistical analysis was conducted using the two-sided Student's *t* test. Unless otherwise stated in figure legends, the biochemical data shown figures in the main text (Figs. 5c, 5d, 7a, 7b, 8a, 8b) or in SI (Fig. S4a−c) are from one representative experiment. For each experiment, 2–3 independent experiments were conducted with similar results.

### Reporting summary
Further information on research design is available in the Nature Portfolio Reporting Summary linked to this article.

## Data availability
The data that support this study are available from the corresponding authors upon reasonable request. The atomic coordinates and structure factors have been deposited in the Protein Data Bank (PDB) with the access code of 8CZJ. The structural data cited in this study are available under accession codes 5TSB, 5TSA and 7Z6N. The source data underlying Figs. 5c, 5d, 7a, 7b, 8a, 8b, 9c and Supplementary Figs. S4a, S8, S12 are provided as a Source Data file. Source data are provided with this paper.

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

## Acknowledgements

The authors thank the beamline scientists at GM/CA-CAT at APS for the assistance in data collections. This work is supported by the NIH grants GM129004 and GM140931 (to J.H.). The work of K.G. and G.W. is supported in part by NIH grants R01GM126189 and R01AI164266 (to G.W).

## Author contributions

J.H. conceived and designed the project. Y.Z., Y.J., J.H., and M.S. conducted structural biology studies, Y.Z., D.S., Y.J., and P.Y. conducted biochemical studies, K.G., G.W., and J.H. conducted computational studies, Y.Z., J.H., G.W., and M.S. analyzed the data. J.H., G.W., M.S., Y.Z., Y.J., and K.G. wrote the manuscript.

## Competing interests

The authors declare no competing interests.
