## [Peer Review File · Nature Communications]

Structural insights into the elevator-type transport mechanism of a bacterial ZIP metal transporterReviewers' Comments:

Reviewer #1:

Remarks to the Author:

The manuscript by Zhang et. al describes a crystal structure of a bacterial ZIP metal transporter in its metal-free form. Compared with an earlier meta-bound form of the same protein, the apo structure reveals a disassembled transport site and 4-helix bundles involved in rigid-body movements. Both the metal-bound and free structures adopt an inward-facing conformation. Authors propose an outward-facing conformation based on two assumptions: (i) an elevator-type motion of the transport site in a 4-helix bundle, (ii) a fixed dimeric interface to support inter-domain motions. Cysteine accessibility and Hg crosslinking assays were used to validate a predicted outward-facing conformation. While the biochemical data are consistent with the structural model, they could also be interpreted by alternative structural models. Overall, the apo structure provides important structural insights into metal binding and release in ZIPs, but it remains unclear whether the binding of metal substrates can trigger a major conformational change as proposed. Authors may consider the following specific points.

1. The proposed dimeric interaction was inconsistent with the apo structure. Since the crosslink data and crystal structure suggest two different forms of dimerization, the dimerization interaction seems weak and variable. What is the biological rationale to use an unstable structure to anchor inter-domain movements?
2. The oligomeric state of the bacterial ZIP is not determined in the native membrane. The proposed elevator-type transport mechanism is based on fixed dimeric interactions. What if the bacterial ZIP protein forms higher oligomeric states in the native membrane?
3. Substrate binding is expected to trigger an alternate exposure of the transport site to either side of the membrane. However, metal-bound and metal-free structures show that metal binding does not change the transport site from inward-facing to outward-facing. Apparently, the existing data do not support an alternate-access scenario. Could an alternative transport mechanism fit the existing structural data better?
4. The rigid body movement of 4-helix bundles is based on a small RMSD within the helix bundle (0.56 Å) as compared to an overall RMSD (1.63 Å) between the metal-bound and metal-free structures. The observed RMSD difference is much smaller in scale than the proposed elevator-type inter-domain motion. Could crystal packing contribute to such a small difference between inter- and intra-domain RMSD?
5. Although the cysteine accessibility data can be explained by the proposed outward-facing conformation, it can be equally explained by many other transport models or dynamic flexibility. Authors should demonstrate a robust change in cysteine accessibility in response to metal binding.
6. N-ethylmaleimide (NEM) is a well-established probe for labeling buried cysteine residues in proteins with high lipid partitioning. Solvent accessibility assays should be performed using a truly hydrophilic thiol reactive probe.
7. The Hg-mediated crosslinking of A95C/L217C was performed in the detergent solution as opposed to the native membrane. Control experiments only validated specific Hg-Cys interactions, but Hg could react with cysteines in denature proteins. More rigorous experiments are required to demonstrate native protein folding before and after Hg treatment. Moreover, unreacted Hg should be removed to prevent secondary crosslinking on SDS-PAGE.
8. The Hg cross-linking data only suggested the proximity of the A95C-L217C pair. Additional data are required to demonstrate metal dependent change in A95C-L217C proximity. To map inter-domain

movements, multiple residue pairs should be surveyed to triangulate relative motions. Again, reciprocal changes should be demonstrated in response to metal binding.

9. Authors should discuss the inherent bias of repeat-swap homology modeling toward the elevator-type transport mechanism.

Reviewer #2:

Remarks to the Author:

The study by Zhang and colleagues in "Structural insights into the elevator-type transport mechanism of a bacterial ZIP metal transporter" is a novel, thorough, and valuable analysis of elevator-type transporter in the Zrt-/Irt-like protein family. Utilizing crystallography, computational analysis, and biochemical assays, the authors have been able describe several additional steps of the transport cycle. They have identified changes in the transporter in going from the outward to inward facing conformations, and upon substrate binding/release. The study is well laid out and thorough. The author's results will prove quite informative and valuable to the transporter field.

However, there are several points which must be addressed in order for this paper to be published.

Major issues

While the authors argue the structural changes within the crystal structure are due to the absence of metal, the low pH of crystallization could also directly lead to conformational changes. Thereby this structure may represent a non-physiological state. The authors should provide evidence that the structure captured is a part of the standard reaction cycle, and not a low pH induced off-pathway structure.

There are significant differences in the oligomer between lipidic cubic phase, native lipid bilayer, and computational models, possibly indicating the dimer is unstable and/or transient. The authors should discuss this possibility, and include calibrated SEC traces from the detergent solubilized samples.

With their biochemical results, the authors have shown changes in proximity and solvent accessibility in agreement with their OFS and conformational change model, and suggest this is a pathway for substrate to access the central binding sites. However, they have not shown this OFS and conformational change are directly linked to substrate transport. Therefore, the authors should perform a transport assay to demonstrate this connection.

Minor issues

While the authors discuss major differences in ternary structure between the OFS and IFS, there may also be side chain rearrangements necessary to this conformational change. Examining their models, these should be noted in the revised text.

The water molecules in figure 2 should be more prominent and clearly labelled.

In lines 131-132 "the formation of a short segment of 310-helix ... Due to the instability of the 310-helix" is confusing and should be revised.

Met99 forms a part of the binding site but is within domain 2. Is there an equivalent residue in the OFS? If not, the author should expound upon the consequences of this structural asymmetry to substrate transport.

The authors collected a quite large and high quality CryoEM dataset, and yet only present images of two 2D classes. A full data processing workflow should be shown in the supplement.

Reviewer #3:

Remarks to the Author:

The manuscript by Zhang et al describes structural characteristics of the bacterial ZIP metal transporter. As the introduction of this paper suggests, while this family of proteins is essential there is much to be learned about the mechanism of this protein. The authors aim to clarify some structural features of this protein. This includes the structure of the apo form of the enzyme as well as confirmation that BbZIP, like other ZIP proteins, is a dimer.

Comments

1. The authors use acidic conditions to solve the full-length structure of BbZIP. Using molecular replacement eases structural determination and it is discovered that the N-terminus co-localizes with the existing 8 TMs. Could this co-localization be a product of the crystallization conditions? Equally, the authors should describe whether this bacterial species encodes any sort of signal sequences. Finally, does this structure with this "9th TM" represent some large conformational change between apo and metal containing structures?

2. Are the cross-linked or variants for cysteine-accessibility proteins functional? At the same time, this reviewer is surprised that no functional data accompanies the authors model of function. The absence of functional data makes the overall model of function appear to be speculative.

Response to reviewers' comments

Note: Reviewers' comments are in blue and our responses are in black

Reviewer #1 (Remarks to the Author):

The manuscript by Zhang et. al describes a crystal structure of a bacterial ZIP metal transporter in its metal-free form. Compared with an earlier meta-bound form of the same protein, the apo structure reveals a disassembled transport site and 4-helix bundles involved in rigid-body movements. Both the metal-bound and free structures adopt an inward-facing conformation. Authors propose an outward-facing conformation based on two assumptions: (i) an elevator-type motion of the transport site in a 4-helix bundle, (ii) a fixed dimeric interface to support inter-domain motions. Cysteine accessibility and Hg crosslinking assays were used to validate a predicted outward-facing conformation. While the biochemical data are consistent with the structural model, they could also be interpreted by alternative structural models. Overall, the apo structure provides important structural insights into metal binding and release in ZIPs, but it remains unclear whether the binding of metal substrates can trigger a major conformational change as proposed. Authors may consider the following specific points.

We thank Reviewer's positive comments on the importance of this work. We believe the new experiments (biochemical and functional studies) presented in the revised manuscript have provided additional data to better support the conclusions.

1. The proposed dimeric interaction was inconsistent with the apo structure. Since the crosslink data and crystal structure suggest two different forms of dimerization, the dimerization interaction seems weak and variable. What is the biological rationale to use an unstable structure to anchor inter-domain movements?

Thanks for asking this important question. Although a previous study showed that BbZIP forms a homodimer in detergent micelles¹, our previous crystal structures solved in lipidic cubic phase only showed monomeric form^{2,3}. It is also the case in our new structure even though the protein was crystallized under a different condition and in a different space group. We had shown that the protein in DDM appears to be a mixture of monomer and dimer based on size estimation in size-exclusion chromatography (Ref2, fig S3). We therefore proposed that the protein in solution is likely in a rapid equilibrium between monomer and dimer, but only the monomeric form is crystallizable under the crystallization conditions. In the revised manuscript, we performed chemical crosslinking experiment of the L138C/M295C variant in the native membrane (*E. coli* membrane fraction) and the result confirms the proposed dimerization mode (**Figure 5d**). We added the following paragraph to discuss the dimerization issue in the revised manuscript. (Page 9)

*“Although oligomerization is very common in elevator transporters, exceptions exist, including the bile acid sodium symporter ASBT with a similar wall-like scaffold domain composed of four TMs as observed in BbZIP^{73,74}. BbZIP indeed forms a dimer in both detergents and in the native membrane (**Figure 5**), but the dimeric form seems to be unstable at least in detergents and in lipidic cubic phase as it has been frequently crystallized in the monomeric form. We have previously shown that the purified protein may be in a rapid monomer-dimer equilibrium in detergents at neutral pH⁴⁶. Oligomerization of the scaffold domain is believed to be beneficial for an elevator transporter’s function, whereas a reversible and potentially tunable oligomerization may allow for regulation. For instance, heterodimerization of ZIP6 and ZIP10 has been shown to be important for their functions to promote cell growth²³. Nevertheless, the transmembrane domain may only partially contribute to dimerization and the extracellular domain (ECD) of some ZIPs, such as the ECD of ZIP4, may play a key role in promoting dimerization for optimal zinc transport⁵⁵.”*

Notably, a recent crystal structure of BbZIP from an independent study (PDB: 7N6D, which was published when this work was under revision⁴. We compared and contrasted the two studies in the last paragraph of the revised manuscript) showed that the protein crystallized at low pH can form two types of crystallographic dimer – one is the same as what we found in our crystal and the other is nearly the same as the dimer model which we proposed in this work. We further discussed the dimerization issue in the last paragraph of the revised manuscript as below. (Page 10)

“Thirdly, the observed loose crystallographic dimer in 7Z6N is not present in our crystal structure (Figure S9c). This difference reinforces the notion that the BbZIP dimer is unstable and the purified protein is likely in a monomer-dimer equilibrium in detergents as we proposed previously⁴⁶. Of great interest, our proposed dimer based on covariance analysis and validated by chemical crosslinking in detergents and in the native membrane is largely consistent with the proposed dimer model based on crystal packing analysis and molecular docking in the other report.”

2. The oligomeric state of the bacterial ZIP is not determined in the native membrane. The proposed elevator-type transport mechanism is based on fixed dimeric interactions. What if the bacterial ZIP protein forms higher oligomeric states in the native membrane?

As discussed above, we have performed chemical crosslinking in the native membrane and confirmed the proposed dimerization mode. It is an open question whether ZIPs form even higher oligomeric states in the native environment. As far as we know, there is no prior report or data supporting this possibility.

3. Substrate binding is expected to trigger an alternate exposure of the transport site to either side of the membrane. However, metal-bound and metal-free structures show that metal binding does not change the transport site from inward-facing to outward-facing. Apparently,

the existing data do not support an alternate-access scenario. Could an alternative transport mechanism fit the existing structural data better?

To support the proposed elevator-type transport mechanism, we have presented the following evidence. *Firstly*, structural comparison between the metal-bound structure and the apo structure showed a rigid body rotation of the four-helix bundle relative to the other 4 TMs, indicating a two-domain architecture which is a shared feature among the elevator transporters. *Secondly*, the evolutionary covariance analysis revealed that the inter-domain interactions are much less conserved than the intra-domain interactions, which provides another piece of evidence supporting the two-domain architecture. *Thirdly*, the OFC model generated by repeat swap homology modeling revealed a vertical rigid body movement of the four-helix bundle (which carries the transport site(s)) relative to the other four TMs. Such a movement is a hallmark of elevator-type transporters. *Fourthly*, the OFC model has been experimentally validated by two complementary biochemical assays. (i) The cysteine accessibility assay conducted in the native membrane suggested the presence of alternative conformation(s) where the residues buried in a hydrophobic environment in the IFC are exposed to a more polar environment. These residues are indeed exposed to the solvent in the OFC model. (ii) More importantly, Hg-mediated chemical crosslinking experiment on multiple residue pairs (as suggested, more residue pairs were tested in the revised manuscript, **Figure 8b**) showed that the four-helix bundle must undergo a large vertical movement to allow crosslinking of the residues not only between $\alpha 2$ (A95 in Domain II) and $\alpha 5$ (L217 in Domain I) but also between $\alpha 4$ (V167 or L169 in Domain I) and $\alpha 7$ (V272 in Domain II). *Fifthly*, chemical crosslinking experiment conducted both in detergents and in the native membrane (new data in the revised manuscript, **Figure 5d**) supported dimerization via Domain II. It is consistent with the common characteristic of elevator transporters that the static scaffold domain is involved in oligomerization. *Lastly*, we identified conserved small residues at the interface between the transport domain and the scaffold domain and systematically replaced them with bulky residues in human ZIP4, a representative and well-characterized ZIP. The results of the cell-based transport assay revealed the importance of these residues for zinc transport, which is consistent with the notion that a smooth domain interface composed of small and hydrophobic residues is crucial for the sliding of the transport domain against the scaffold domain for an elevator transporter.

Among the known major transport mechanisms (rocker switch, rocking bundle, and elevator⁵), only the elevator mode can explain all the results listed above. The transporters using the rocker switch mechanism consist of two structurally similar domains which rock around each other to achieve alternating access. In contrast, the two domains of BbZIP are totally different in structure. The rocking bundle and elevator mechanisms are more similar but one prominent difference between them is that the transport site of the former is formed by residues from both dissimilar domains whereas the transport site of the latter is (nearly) exclusively composed of the residues from one domain (transport domain). This critical difference allows the elevator transporters to translocate the bound substrate across the membrane when the

transport domain moves vertically relative to the static scaffold domain. In BbZIP, the binuclear metal center (transport sites) is composed of the residues from $\alpha 4$, $\alpha 5$, and $\alpha 6$, all of which are in Domain I (transport domain). One residue (M99) is from $\alpha 2$ (Domain II), but it is not always coordinated with substrate (PDB: 5TSA) and less conserved (not present in human ZIPs, for instance). This structural feature makes BbZIP a likely candidate of elevator-type transporter. Importantly, when the tested residue pairs are crosslinked by Hg^{2+} , the transport site must be concomitantly moved upward toward the extracellular side, which is a common characteristic of elevator-type transporters. To test whether BbZIP may hypothetically use a rocking bundle-like mechanism, we generated an OFC model where Domain I is allowed to rock around Domain II while the transport site is maintained at the same place. As shown in Fig. 1 below, when Domain I is rotated to reach the limit (without clashing with Domain II), the distance between the C_β atoms of A95 and L217 was only shortened by 1.2 Å (from 13.3 Å to 12.1 Å), which is still much longer than the maximum distance allowing for Hg-mediated crosslinking (7.8 Å). With the above considerations, we concluded that BbZIP is an elevator transporter. Honestly, we cannot exclude the possibility that BbZIP may utilize a completely new and unprecedented mechanism since there is no experimentally determined OFC structure at this moment. With this consideration, we added the following sentence in the revised manuscript. (Page 10)

"..., but we cannot completely exclude the possibility that BbZIP may use an unprecedented transport mechanism in the absence of an experimentally determined structure representing the OFC state."

4. The rigid body movement of 4-helix bundles is based on a small RMSD within the helix bundle (0.56 Å) as compared to an overall RMSD (1.63 Å) between the metal-bound and metal-free structures. The observed RMSD difference is much smaller in scale than the proposed elevator-type inter-domain motion. Could crystal packing contribute to such a small difference between inter- and intra-domain RMSD?

We would like to clarify that the structure solved in this work is a still an IFC but in a metal free state. When this structure is compared with the previously solved IFC in the metal-bound, the resulting RMSD is therefore much smaller than that obtained through comparison of an IFC with an OFC for a typical elevator transporter.

A detailed crystal packing is shown in Fig. 2 below. Packing analysis shows that the asymmetric units associates each other through the interactions between the N-terminal domains and the scaffold domains. Only one of $\alpha 4$ s in the crystallographic dimer is involved in packing. Therefore, the rigid body rotation of the transport domain relative to the scaffold domain is unlikely caused by the packing induced structural distortion. Another evidence supporting this conclusion comes from the newly reported BbZIP crystal structure with a different space group (PDB: 7Z6D, C2221 vs P21 of our structure), where a similar but a different rotation of the same 4-TM bundle relative to the other TMs was observed (**Figure S9a**). Therefore, it is unlikely that the rigid body rotation of the 4-TM bundle is a crystal packing artifact.

5. Although the cysteine accessibility data can be explained by the proposed outward-facing conformation, it can be equally explained by many other transport models or dynamic flexibility. Authors should demonstrate a robust change in cysteine accessibility in response to metal binding.

Thanks for the suggestion. To examine the influence of metal binding on the IFC-OFC equilibrium of BbZIP, we conducted cysteine accessibility assay on two variants, A203C and L200C in which the cysteine residues are exposed only in the OFC, in the presence or absence of zinc ions. As shown in **Figure 7b**, although the overall profiles of the dose-dependent NEM labeling are similar with and without the added zinc ions (which may suggest that zinc binding does not drastically change the IFC-OFC equilibrium), a close inspection of the data revealed that, at the transition point of the NEM titration (200 μ M of NEM for A203C and 100 μ M of NEM for L200C), the accessibility of both residues to NEM was reduced, which was indicated by the higher percentage of mPEG5k labeling. This result suggests that zinc binding to BbZIP favors the IFC, although only modestly, over the OFC, which is consistent with proposed mechanism illustrated in **Figure 10**. Due to the relative low sensitivity and the nature of a population study, the current cysteine accessibility assay is unable to provide more details about metal binding/release triggered conformational change and altered dynamics. Other approaches, particularly those at the single molecule level, may be better choices for this type of the research. This is an important topic for future study but out of the scope of this work.

6. N-ethylmaleimide (NEM) is a well-established probe for labeling buried cysteine residues in proteins with high lipid partitioning. Solvent accessibility assays should be performed using a truly hydrophilic thiol reactive probe.

As far as we know, NEM has been broadly applied to assess accessibility of cysteine residues in membrane proteins ⁶. Modifying a cysteine residue with NEM requires both of the following conditions: (1) The reactivity of the cysteine needs to be high enough to react with maleimide, which is governed by local hydrophilicity ⁷. As stated in Ref 7, "Maleimides react with thiolate anions, which require water exposure. They will not react with Cys residues in the low-dielectric (i.e., lipid exposed) membrane environment, because the thiol group is protonated and unreactive". In our understanding, the reactivity of a cysteine residue in a fully hydrophobic environment would be much lower when compared to those more exposed to solvent. As shown in **Figure 7a**, among the NEM-modifiable residues (A184C, L200C, A203C, L92C, and A214C), none was found to have a particularly low reactivity with NEM when compared to each other (as indicated by nearly no mPEG5k labeling after NEM treatment), suggesting that the cysteine residues at these positions are exposed to the environment with similar polarity. For L92C and A214C, the result is consistent with the IFC where they are exposed to solvent. For A184C, L200C and A203C, the result suggests the presence of alternative conformations where the cysteine residues at these positions are exposed to solvent as much as L92C and A214C. (2) The tertiary structure allows for NEM reaching the cysteine residue. When a cysteine residue is buried in protein core and tightly packed with other residues, its accessibility to NEM would be much lower than those exposed to an open space (either the external bulky solvent or an internal cavity of the protein of interest). Indeed, as shown in **Figure 7a**, three residues in the middle of the transport pathway (A102C, Q207C, and V272C) were unable to be labeled by NEM, which is consistent with the structure (or the model) where they are always tightly packed with surrounding residues. In contrast, the other

tested residues were similarly modified by NEM. For L92C and A214C, as they are exposed to an open space in the IFC, it is reasonable that they are readily modified by NEM. For A184C, L200C and A203C, their modification by NEM strongly suggest that they are similarly exposed to an open space which NEM can enter and react with the cysteine residues. Collectively, the data suggest that (1) A184C, L200C, and A203C are exposed to an open space in an alternative conformation(s) (to allow NEM to approach them) and (2) in this alternative conformation(s), the local environment of the cysteine residues in A184C, L200C, and A203C is as hydrophilic as the local environment in L92C and A214C (to allow cysteine to react with NEM). With these considerations, we believe the current data can support our claim that there must be an alternative conformation(s) to allow A184C, L200C, and A203C to at least transiently be exposed a hydrophilic environment.

To further confirm cysteine accessibility, we tested mPEG5k labeling to the variants of L92C, L200C, A203C, and A214C by using a protocol similar to a recent report⁸. In this experiment, the purified protein in DDM was allowed to react with 1 mM mPEG5k for 0 or 30 minutes and the reaction was terminated by adding 100 mM water soluble thiol reacting reagent methyl methanethiosulfonate. Due to the long and hydrophilic tail, mPEG5k can only approach and react with a thiol group in a highly hydrophilic environment. As shown in **Figure S4a**, all the tested variants, including those exposed to the solvent only in the OFC model, were PEGylated by mPEG5k. We added the following sentences in the revised manuscript. (Page 6)

"When the long and highly hydrophilic mPEG5K molecules were directly applied to the purified variants (L92C, L200C, A203C, and A214C), cysteine PEGylation occurred for all the tested variants at comparable levels (Figure S4a), confirming that these residues are indeed similarly exposed to an aqueous environment."

7. The Hg-mediated crosslinking of A95C/L217C was performed in the detergent solution as opposed to the native membrane. Control experiments only validated specific Hg-Cys interactions, but Hg could react with cysteines in denature proteins. More rigorous experiments are required to demonstrate native protein folding before and after Hg treatment. Moreover, unreacted Hg should be removed to prevent secondary crosslinking on SDS-PAGE.

As described in Materials and Methods, after the reaction with Hg (at tens of micromolar), the samples were treated with 2 mM NEM to block any unreacted cysteine residues. It is unlikely that a significant level of Hg-mediated crosslinking occurred under denatured conditions in the presence of high concentration of NEM. To further confirm this, we applied the Hg-crosslinked A95C/L217C to a size-exclusion column, which had been equilibrated with a buffer without Hg, to remove any free Hg in the sample. As shown in **Figure S4c**, the gel filtration profile showed no sign of aggregation or higher degree of oligomerization. Importantly, the Hg-mediated crosslinking is maintained after size-exclusion chromatography (as indicated by the band shift in SDS-PAGE, **Figure S4c**), further confirming that the crosslinking occurred when protein is in the native state, not under a denaturing condition. As suggested by

Reviewer, we also conducted the crosslinking experiment in *E.coli* membrane fraction, and the result shows that crosslinking occurs when the A95C/L217C variant is in its native environment (**Figure S4b**).

8. The Hg cross-linking data only suggested the proximity of the A95C-L217C pair. Additional data are required to demonstrate metal dependent change in A95C-L217C proximity. To map inter-domain movements, multiple residue pairs should be surveyed to triangulate relative motions. Again, reciprocal changes should be demonstrated in response to metal binding.

Thanks for the suggestions. We had planned to test metal dependent crosslinking, but later realized that this experiment could be very difficult to conduct. The major issue is that Hg-mediated crosslinking reaction completes within minutes⁹, and once the Hg-S covalent bonds are formed, the protein will be locked in a fixed state permanently. Because of the transporter's rapid equilibrium among distinct conformational states and the rapid irreversible Hg-mediated crosslinking reaction, the percentage of crosslinking would only reflect the Hg/protein molar ratio, rather than the equilibrium among IFC, OFC, and other intermediate states. Therefore, we are afraid that the suggested experiment may not provide useful insights into metal dependent conformational change. However, by following the Reviewer's suggestion, we have shown in the new NEM labeling experiments that zinc binding to the transporter modestly favors IFC (**Figure 7b**).

To further validate the proposed OFC model, we conducted Hg-mediated crosslinking on additional three variants (V167C/V272C, W168C/V272C, and L169C/V272C) (**Figure 8b**). V167, W168 and L169 are in $\alpha 4$ and V272 is in $\alpha 7$. These residues are all distal from A95 (in $\alpha 2$) and L217 (in $\alpha 5$). As shown in **Figure 8b**, two variants (167C/272C and 169C/272C) were crosslinked by Hg with band shift in SDS-PAGE, whereas the 168C/272C variant was not under the same condition. In the proposed OFC model, residue 167 and residue 169 are at least partially facing toward residue 272, whereas residue 168 is located on the side of $\alpha 4$ pointing away from residue 272. This result provides new evidence supporting the proposed OFC model, and also indicates that the Hg-mediated crosslinking is specific.

9. Authors should discuss the inherent bias of repeat-swap homology modeling toward the elevator-type transport mechanism.

We agree that repeat-swap homology modeling, as other modeling approaches, may cause artifact. In our practice, repeat-swap homology modeling can almost for sure generate a conformational different from the original structure if symmetry is used correctly. Then, one must examine whether or not the generated model reveals, through structural comparison, the hallmark of an elevator transporter – a vertical rigid body movement of the transport domain, which exclusively (or almost exclusively) harbors the transport site, relative to the static scaffold domain during the IFC-OFC interconversion. Importantly, the generated model must be experimentally validated to exclude possible artifact. For this reason, we conducted

two complementary biochemical assays to examine the proposed OFC model. Based on the validated OFC model as well as other evidence presented in this work, we are confident that BbZIP uses the elevator-type transport mechanism.

In the revised manuscript, we added the following sentence on page 6:

"Next, we experimentally examined the computationally generated OFC model to exclude the potential bias of repeat-swap homology modeling toward elevator-like transport mechanism."

Reviewer #2 (Remarks to the Author):

The study by Zhang and colleagues in "Structural insights into the elevator-type transport mechanism of a bacterial ZIP metal transporter" is a novel, thorough, and valuable analysis of elevator-type transporter in the Zrt-/Irt-like protein family. Utilizing crystallography, computational analysis, and biochemical assays, the authors have been able describe several additional steps of the transport cycle. They have identified changes in the transporter in going from the outward to inward facing conformations, and upon substrate binding/release. The study is well laid out and thorough. The author's results will prove quite informative and valuable to the transporter field.

We appreciate Reviewer's positive comments on our work.

However, there are several points which must be addressed in order for this paper to be published.

Major issues

1. While the authors argue the structural changes within the crystal structure are due to the absence of metal, the low pH of crystallization could also directly lead to conformational changes. Thereby this structure may represent a non-physiological state. The authors should provide evidence that the structure captured is a part of the standard reaction cycle, and not a low pH induced off-pathway structure.

Thanks for bringing up this important issue, which we have been thinking about for a while. Below are our arguments on this issue. *Firstly*, as reported by Dax Fu's group ¹, BbZIP exhibits the highest transport activity at pH 5, which was the lowest pH tested in their study. In the same study, it was shown that BbZIP behaves well in size-exclusion chromatography in a buffer at pH 4. As a matter of fact, ZIP functioning at low pH is not rare. For example, ZIP4, which transports zinc ions in small intestine, particularly at duodenum where the pH is 6 and below ¹⁰. *Secondly*, one of the major findings from this structure is the rigid body movement of the 4-helix bundle (transport domain) relative to the other four TMs (scaffold domain), a hallmark of elevator transporters. Like many other elevator transporters, small and

hydrophobic residues are dominant at the interface of the two domains. As these residues are non-polar, lowering pH should not lead to any significant structural changes for these residues. *Thirdly*, another major finding from the new structure is that, when the transporter is in the metal-free state, the primary transport site (M1) is totally disassembled, whereas the secondary transport site (M2) is still well maintained. As several metal chelating residues (H177, E181, D208, E211, and E240) at the transport site can be protonated, their structures may change at low pH. Given that protons and metal ions are Lewis acids, one would expect that protons (or hydroniums) play the same role as metal ions in maintaining the metal-bound conformation. In contrast, M1 is disassembled, indicating that the structural changes caused by the loss of metal substrate at M1 cannot be compensated by proton binding to the same site. Collectively, we believe that the significant structural changes observed in the new metal-free structure are unlikely to be artifacts caused by low pH.

2. There are significant differences in the oligomer between lipidic cubic phase, native lipid bilayer, and computational models, possibly indicating the dimer is unstable and/or transient. The authors should discuss this possibility, and include calibrated SEC traces from the detergent solubilized samples.

Thanks for pointing out this important issue. As a response to Reviewer#1's request, we conducted the chemical crosslinking experiment on the L138C/M295C variant in the native membrane (membrane fraction), and the result confirmed this dimerization mode (**Figure 5d**). As suggested, we added the following paragraph to discuss the dimerization issue. (Page 9)

"Although oligomerization is very common in elevator transporters, exceptions exist, including the bile acid sodium symporter ASBT with a similar wall-like scaffold domain composed of four TMs as observed in BbZIP^{73,74}. BbZIP indeed forms a dimer in both detergents and in the native membrane (Figure 5), but the dimeric form seems to be unstable at least in detergents and in lipidic cubic phase as it has been frequently crystallized in the monomeric form. We have previously shown that the purified protein may be in a rapid monomer-dimer equilibrium in detergents at neutral pH⁴⁶. Oligomerization of the scaffold domain is believed to be beneficial for an elevator transporter's function, whereas a reversible and potentially tunable oligomerization may allow for regulation. For instance, heterodimerization of ZIP6 and ZIP10 has been shown to be important for their functions to promote cell growth²³. Nevertheless, the transmembrane domain may only partially contribute to dimerization and the extracellular domain (ECD) of some ZIPs, such as the ECD of ZIP4, may play a key role in promoting dimerization for optimal zinc transport⁵⁵."

Another piece of evidence supporting the proposed dimerization model comes from a new BbZIP structure (PDB 7Z6N, recently published while this work is under revision⁴. We compared and contrasted the two studies in the last paragraph of the revised manuscript.). That structure shows two types of crystallographic dimer – one is the same as we observed in our structure and the other is consistent with the dimer we proposed in this work. We further

discussed the dimerization issue in the last paragraph of the revised manuscript as below. (Page 10)

“Thirdly, the observed loose crystallographic dimer in 7Z6N is not present in our crystal structure (Figure S9C). This difference reinforces the notion that the BbZIP dimer is unstable and the purified protein is likely in a monomer-dimer equilibrium in detergents as we proposed previously⁴⁶. Of great interest, our proposed dimer based on covariance analysis and validated by chemical crosslinking in detergents and in the native membrane is largely consistent with the proposed dimer model based on crystal packing analysis and molecular docking in the other report.”

As for the suggested SEC experiment, we previously reported a calibrated SEC profile of purified BbZIP in DDM². The figure below showed that there seems to be a monomer-dimer equilibrium in DDM. As a matter of fact, both species (the two peaks in SEC profile) were crystallized in our previous study but there was no difference in structure. The SEC profiles are not always consistent from batch to batch, against suggesting that the monomer-dimer equilibrium is variable.

3. With their biochemical results, the authors have shown changes in proximity and solvent accessibility in agreement with their OFS and conformational change model, and suggest this is a pathway for substrate to access the central binding sites. However, they have not shown this OFS and conformational change are directly linked to substrate transport. Therefore, the authors should perform a transport assay to demonstrate this connection.

Thanks for the great suggestion. We agree that functional study is important to exam the proposed transport mechanism. Compared with transporters utilizing other transport modes,

the most differential characteristic of an elevator transporter is the rigid body sliding of the transport domain against the scaffold domain during substrate transport, which is facilitated by a smooth domain-domain interface. Through sequence alignment, we identified multiple small residues (Ala, Gly, or Ser) which are located at the proposed domain interface and conserved in representative ZIPs from major subfamilies. As these residues are highly conserved, they are supposed to exert the same or similar functions in the fundamental transport mechanism shared by the entire family. To experimentally examine the importance of these small residues, we replaced them individually with bulky amino acids (Val and Phe) in human ZIP4 and then examined the zinc transport activities of the variants using the well-established cell-based radioactive metal transport assay which my group has been conducted since 2016. The results are shown in **Figure 9** and the following new section was added in the revised manuscript. (Page 7)

"Crucial roles of small residues at the domain interface

The hallmark of elevator-type transporters is the rigid body sliding of the transport domain, which exclusively (or almost exclusively) carries substrate(s), against the static scaffold domain. This movement is facilitated by a smooth and generally hydrophobic interface between the two domains. For example, a "greasy" domain interface consisting primarily of small residues was observed in CitS, a Na⁺/citrate symporter and an established elevator transporter⁶⁶. Similarly, structural inspection revealed that, out of approximately forty residues at the interface between the two domains of BbZIP, nearly half of them are small residues (Ala, Gly, or Ser) (Figure S5). Multiple sequence alignment of ZIPs from different subfamilies identified four highly conserved small residues (A95, A184, A203, and A214 in BbZIP, Figures 9A, B) at the domain interface. To examine the importance of these residues for zinc transport, we individually substituted each of the corresponding residues with a valine (hydrophobic with a slightly bigger sidechain) and a phenylalanine (hydrophobic with a bulky and rigid sidechain) in human ZIP4 (A386, A514, A532, and G543), a well-characterized ZIP for which the cell-based zinc transport assay has been frequently conducted in recent studies^{34, 48, 55, 67, 68, 69}. As shown in Figure 9C, substitution of A386 and A532 with valine greatly reduced the zinc transport activity by more than 80% and substitution of A532 with phenylalanine completely eliminated transport activity. Substitution of A514 and G543 with valine also significantly reduced activity but to a lesser extent. Phenylalanine substitution of A514 led to a greater activity suppression, but the G543F variant unexpectedly exhibited an activity similar to the wild type ZIP4. Compared with A386 and A532, A514 and G543 are at the peripheral region of the domain interface (Figure 9B), which may explain why substitution with bulky amino acids at these two positions caused less disruption of transporter's function. The same pattern was also observed when these residues were replaced with cysteine residues (Figure S8). Overall, these results demonstrated the importance of the conserved small residues at the domain interface and supported the notion that a smooth domain interface is required for optimal activity of an elevator transporter, which is in line with the previous reports that mutations at the domain interface of elevator transporters may drastically influence transport activity^{70, 71, 72}."

Minor issues

1. While the authors discuss major differences in ternary structure between the OFS and IFS, there may also be side chain rearrangements necessary to this conformational change. Examining their models, these should be noted in the revised text.

Thanks for the suggestion. We examined the model to identify polar interactions only present in the OFC. This survey led to the identification of the interactions between R166 in $\alpha 4$ in the transport domain and two residues (H275 and E276) in $\alpha 7$ in the scaffold domain. As shown in Figure S7, the hydrogen bonds formed between these residues may stabilize the OFC and they may also function as a cytoplasmic gate to prevent metal leak. We discussed this putative gate function in the revised manuscript as below. (Page 9)

"One putative candidate for the gate at the cytoplasmic side consists of R166 in $\alpha 4$ of the transport domain and two residues (H275 and E276) in $\alpha 7$ of the scaffold domain (Figure S7). In the IFC, R166 is distant from H275 and E276, both of which coordinate metal substrates in the release pathway; in the OFC model, R166 approaches the two metal chelating residues to form hydrogen bonds which may stabilize the OFC and contribute to block the transport pathway to prevent leak."

2. The water molecules in figure 2 should be more prominent and clearly labelled.

The figure has been updated with enlarged and more clearly labeled water molecules.

3. In lines 131-132 "the formation of a short segment of 310-helix ... Due to the instability of the 310-helix" is confusing and should be revised.

This sentence has been revised as "..., leading to the formation of a short 3_{10} -helix (spanning residues 176-179) at the bending point of $\alpha 4$. As a result, ..."

4. Met99 forms a part of the binding site but is within domain 2. Is there an equivalent residue in the OFS? If not, the author should expound upon the consequences of this structural asymmetry to substrate transport.

Thanks for this good question. A potential metal chelating residue from Domain II is S106 which is located at the pore entrance and forms a putative metal binding site in the OFC (**Figure 6c**). We added the following sentence on page 6.

"M99 in $\alpha 2$ may play a role in stabilizing the IFC, and this function in the OFC may be replaced by S106 in $\alpha 2$ which appears to participate in metal chelation at the extracellular side."

5. The authors collected a quite large and high quality CryoEM dataset, and yet only present images of two 2D classes. A full data processing workflow should be shown in the supplement.

As suggested, we have added a brief flowchart to show the procedure of data processing in SI (**Figure S10**).

Reviewer #3 (Remarks to the Author):

The manuscript by Zhang et al describes structural characteristics of the bacterial ZIP metal transporter. As the introduction of this paper suggests, while this family of proteins is essential there is much to be learned about the mechanism of this protein. The authors aim to clarify some structural features of this protein. This includes the structure of the apo form of the enzyme as well as confirmation that BbZIP, like other ZIP proteins, is a dimer.

Thanks for Reviewer's nice summary of our work.

Comments

1. The authors use acidic conditions to solve the full-length structure of BbZIP. Using molecular replacement eases structural determination and it is discovered that the N-terminus co-localizes with the existing 8 TMs. Could this co-localization be a product of the crystallization conditions? Equally, the authors should describe whether this bacterial species encodes any sort of signal sequences. Finally, does this structure with this "9th TM" represent some large conformational change between apo and metal containing structures?

Thanks for the questions about the extra TM. As shown in **Figure S1**, the amino acid sequence of the N-terminal segment (residue 23-49) is highly hydrophobic and predicted to form a transmembrane helix by TMHMM (see Fig. 3 below). We also examined whether this segment in Gram negative bacteria (BbZIP is from *Bordetella bronchiseptica*) could be a signal peptide using SignalP 5.0, and the result suggested that the likelihood of this segment being a signal peptide is less than 5%. As $\alpha 0$ (and together with the N-terminal amphipathic helix $\alpha 0a$) associates with $\alpha 3/4/6$ and there are significant differences between the structures of the metal-free state and the metal bound state, it is possible that the conformation of the metal-free state better stabilizes this previously severely disordered segment. Consistently, a recently reported BbZIP structure (PDB 7Z6N, published while this work is under revision⁴). We compared and contrasted the two studies in the last paragraph of the revised manuscript) also revealed this extra TM although the space group of that structure is different from that of our structure. Therefore, it is unlikely that the structure of $\alpha 0$ is a crystallographic artifact. We added the following sentence in the revised manuscript. (Page 3)

"The significantly changed structure of the eight-TM core (discussed later) may better stabilize this otherwise highly flexible segment."

2. Are the cross-linked or variants for cysteine-accessibility proteins functional? At the same time, this reviewer is surprised that no functional data accompanies the authors model of function. The absence of functional data makes the overall model of function appear to be speculative.

Thanks for the great suggestion. We agree that functional study is important to exam the proposed transport mechanism.

Functional study to examine transport mechanism

Compared with transporters utilizing other transport modes, the most differential characteristic of an elevator transporter is the rigid body sliding of the transport domain against the scaffold domain during substrate transport, which is facilitated by a smooth domain-domain interface. Through sequence alignment, we identified multiple small residues (Ala, Gly, or Ser) which are located at the proposed domain interface and conserved in representative ZIPs from major subfamilies. As these residues are highly conserved, they are supposed to exert the same or similar functions in the fundamental transport mechanism shared by family members. To experimentally examine the importance of these small residues, we replaced them individually with bulky amino acids (Val and Phe) in human ZIP4 and then examined the zinc transport activities of the variants using the well-established cell-based radioactive metal transport assay which my group has been conducted in the last several years. The results are shown in **Figure 9** and the following new section was added in the revised manuscript. (Page 7)

"Crucial roles of small residues at the domain interface

The hallmark of elevator-type transporters is the rigid body sliding of the transport domain, which exclusively (or almost exclusively) carries substrate(s), against the static scaffold domain.

This movement is facilitated by a smooth and generally hydrophobic interface between the two domains. For example, a “greasy” domain interface consisting primarily of small residues was observed in CitS, a Na⁺/citrate symporter and an established elevator transporter⁶⁶. Similarly, structural inspection revealed that, out of approximately forty residues at the interface between the two domains of BbZIP, nearly half of them are small residues (Ala, Gly, or Ser) (Figure S5). Multiple sequence alignment of ZIPs from different subfamilies identified four highly conserved small residues (A95, A184, A203, and A214 in BbZIP, Figures 9A, B) at the domain interface. To examine the importance of these residues for zinc transport, we individually substituted each of the corresponding residues with a valine (hydrophobic with a slightly bigger sidechain) and a phenylalanine (hydrophobic with a bulky and rigid sidechain) in human ZIP4 (A386, A514, A532, and G543), a well-characterized ZIP for which the cell-based zinc transport assay has been frequently conducted in recent studies^{34, 48, 55, 67, 68, 69}. As shown in Figure 9C, substitution of A386 and A532 with valine greatly reduced the zinc transport activity by more than 80% and substitution of A532 with phenylalanine completely eliminated transport activity. Substitution of A514 and G543 with valine also significantly reduced activity but to a lesser extent. Phenylalanine substitution of A514 led to a greater activity suppression, but the G543F variant unexpectedly exhibited an activity similar to the wild type ZIP4. Compared with A386 and A532, A514 and G543 are at the peripheral region of the domain interface (Figure 9B), which may explain why substitution with bulky amino acids at these two positions caused less disruption of transporter’s function. The same pattern was also observed when these residues were replaced with cysteine residues (Figure S8). Overall, these results demonstrated the importance of the conserved small residues at the domain interface and supported the notion that a smooth domain interface is required for optimal activity of an elevator transporter, which is in line with the previous reports that mutations at the domain interface of elevator transporters may drastically influence transport activity^{70, 71, 72}.”

Functional study to examine the cysteine variants’ activity

Among the 15 residues which were mutated to cysteine in the revised manuscript, 10 of them (L92, A102, L138, V167, W168, L169, L200, L217, V272, and M295) are highly variable in ZIPs (Fig. 4, next page), so cysteine replacement at these positions is unlikely to completely eliminate the transporter’s activity. One residue (Q207) is within the primary transport site, but mutation of the corresponding residue in ZIP4 (H536A) only modestly reduced activity by 50%². We then focused on the other four conserved residues (A95, A184, A203, and A214), substituted the corresponding residues with cysteine in ZIP4 and tested transport activity of the ZIP4 variants. As shown in **Figure S8**, the tested cysteine variants are functional but with significantly reduced activity (by 40-60%), which again indicates the importance of the small residues at the domain interface (**Figure 9**).

For the Hg-crosslinked variant (A95C/L217C), although we expect that it loses transport activity as it is locked in the OFC, we couldn't find a proper experimental approach to evaluate the activity due to following technical concerns. *Firstly*, A95 and L217 are either buried (in the OFC) or face toward cytoplasm (in the IFC). If we choose to use the cell-based transport assay to study the Hg-crosslinked variant (of ZIP4), we have to expect that enough Hg²⁺ applied to cells can pass through cell membrane without affecting cell viability. However, we think it would be difficult because of the high toxicity of Hg to mammalian cells. In addition, as there are 15 cysteine residues in ZIP4 (eight in extracellular domain and seven in transmembrane domain), Hg may crosslink cysteine residues in a non-predictable manner, resulting in uninterpretable data. *Secondly*, if we opt to examine the activity using proteoliposome-based assay, we will need to encapsulate a sensitive zinc fluorescence probe, such as the frequently used FluoZin-3, at a high concentration (for instance, 200 μM in the study of ZnT8¹¹) in proteoliposome. As FluoZin-3 is a strong chelator of Hg²⁺ (see Fig. 5 below), it is possible that FluoZin-3 will partially or completely deprive Hg²⁺ from protein and breaks the linkage during

sample preparation. Nevertheless, the gel filtration profile of the crosslinked A95C/L217C variant showed that the protein is well folded and stable in DDM after crosslinking (**Figure S4c**).

References:

1. Lin W, Chai J, Love J, Fu D. Selective electrodiffusion of zinc ions in a Zrt-, Irt-like protein, ZIPB. *J Biol Chem* **285**, 39013-39020 (2010).
2. Zhang T, Liu J, Fellner M, Zhang C, Sui D, Hu J. Crystal structures of a ZIP zinc transporter reveal a binuclear metal center in the transport pathway. *Sci Adv* **3**, e1700344 (2017).
3. Zhang T, Sui D, Zhang C, Cole L, Hu J. Asymmetric functions of a binuclear metal center within the transport pathway of a human zinc transporter ZIP4. *FASEB J* **34**, 237-247 (2020).
4. Wiuf A, *et al.* The two-domain elevator-type mechanism of zinc-transporting ZIP proteins. *Sci Adv* **8**, eabn4331 (2022).
5. Drew D, Boudker O. Shared Molecular Mechanisms of Membrane Transporters. *Annu Rev Biochem* **85**, 543-572 (2016).
6. Guan L, Kaback HR. Site-directed alkylation of cysteine to test solvent accessibility of membrane proteins. *Nat Protoc* **2**, 2012-2017 (2007).
7. Chen Y, *et al.* YidC Insertase of Escherichia coli: Water Accessibility and Membrane Shaping. *Structure* **25**, 1403-1414 e1403 (2017).
8. Sampson CDD, Stewart MJ, Mindell JA, Mulligan C. Solvent accessibility changes in a Na(+)-dependent C4-dicarboxylate transporter suggest differential substrate effects in a multistep mechanism. *J Biol Chem* **295**, 18524-18538 (2020).
9. Ren Z, *et al.* Structure of an EIIIC sugar transporter trapped in an inward-facing conformation. *Proc Natl Acad Sci U S A* **115**, 5962-5967 (2018).
10. Evans DF, Pye G, Bramley R, Clark AG, Dyson TJ, Hardcastle JD. Measurement of gastrointestinal pH profiles in normal ambulant human subjects. *Gut* **29**, 1035-1041 (1988).
11. Merriman C, Li H, Li H, Fu D. Highly specific monoclonal antibodies for allosteric inhibition and immunodetection of the human pancreatic zinc transporter ZnT8. *J Biol Chem* **293**, 16206-16216 (2018).

Reviewers' Comments:

Reviewer #1:

Remarks to the Author:

The authors did a good job validating the proposed transport mechanism. This work provides important insights into a role of dimerization in driving transmembrane zinc crossing in ZIPs. The paper is well written and supported by detailed structural and functional analysis. No further concern.

This work shows that low pH is required to stabilize the apo structure but is not able to drive a major conformational switch from IFC to OFC. Early kinetic study of BbZIP showed that the zinc transport activity is pH dependent, but zinc flux is not driven by the transmembrane gradient.

Reviewer #2:

Remarks to the Author:

The study by Zhang and colleagues in "Structural insights into the elevator-type transport mechanism of a bacterial ZIP metal transporter" is a novel, thorough, and valuable analysis of elevator-type transporter in the Zrt-/Irt-like protein family. Utilizing crystallography, computational analysis, and biochemical assays, the authors have been able describe several additional steps of the transport cycle. They have identified changes in the transporter in going from the outward to inward facing conformations, and upon substrate binding/release. The study is well laid out and thorough.

The author's results will prove quite informative and valuable to the transporter field. In this revision, the authors have also adequately addressed the issues raised in my review.

Reviewer #3:

Remarks to the Author:

The authors have made substantial revisions to the manuscript. A couple of remaining queries remain:

1. This reviewer previously inquired about signal sequences. The authors provided additional information about the possibility of a signal sequence in their response, but it is not evident that this has been added in the last paragraph of the manuscript as the authors indicate. It would also be appropriate to add Figure 3 of the response into the supplemental data with the associated text.
2. The authors agree that measuring cross-linked or variants for cysteine accessibility proteins function is important. However, the authors decided to measure only four of fifteen cysteine variants. The authors mention Q207, but this reviewer does not see that residue in Fig. 4 of the response. Is there functional data from a previous paper that could be described here. Neither is H536 shown. By focusing only on four residues, the authors examine only a subset. Just because residues are not conserved does not mean that they don't have a role. Therefore these cys residues and the experiments should be expanded beyond the four targeted in this initial re-submission.
3. The authors raise an interesting issue with the problem of measuring protein function after mercury-crosslinking. Naturally, creating a Cys-less functional version of the transporter would ameliorate the issues the authors mention. Presumably, the authors do not want to attempt this. An alternative is not to use Hg cross-linking, but another cross-linking approach. Finally, is Hg transported by this protein?

Reviewer #4:

Remarks to the Author:

The manuscript by Zhang et al., is a neat work that uses crystallography to obtain the apo-form of a bacterial ZIP metal transporter. The structure is resolved in the inward facing conformation (as the one resolved previously with the metal bound). They analyzed the possible dimerization of the ZIP transporter, suggesting that the dimer is unstable. Analyzing evolutionary couplings (and AlphaFold multimer) they suggest a dimer interface, validating it with cysteine crosslink. Finally, they generated an outward facing conformation using repeat swap homology modeling from the inward facing structure and validated this model with biochemistry assays. In this present form the work is interesting, the results seem robust, and it will certainly be of relevance in the field of membrane transporters. I have only few minor points.

1. The author should provide the details of the AlphaFold modeling. How many structures did they generate? Which database they used?
2. The authors may provide the full alignments for the repeat swap model or in the SI or in a database entry (i.e. Zenodo)
3. I do not agree with the statement that the repeat swap modeling has a bias toward the elevator mechanism. The main assumption of this approach is the definition of the repeats. Indeed, the same approach has been used to predict different mechanism, and these predictions depended only in the asymmetry between the repeats. Thus, the predicted outward facing model is another, not biased, evidence that supports the elevator mechanism.

Response to Review

Reviewers' comments in black and our response in blue.

Reviewer #1 (Remarks to the Author):

The authors did a good job validating the proposed transport mechanism. This work provides important insights into a role of dimerization in driving transmembrane zinc crossing in ZIPs. The paper is well written and supported by detailed structural and functional analysis. No further concern.

This work shows that low pH is required to stabilize the apo structure but is not able to drive a major conformational switch from IFC to OFC. Early kinetic study of BbZIP showed that the zinc transport activity is pH dependent, but zinc flux is not driven by the transmembrane gradient.

Thanks for the positive comments. We agree on the comments about the effects of pH on protein structure and function.

Reviewer #2 (Remarks to the Author):

The study by Zhang and colleagues in "Structural insights into the elevator-type transport mechanism of a bacterial ZIP metal transporter" is a novel, thorough, and valuable analysis of elevator-type transporter in the Zrt-/Irt-like protein family. Utilizing crystallography, computational analysis, and biochemical assays, the authors have been able describe several additional steps of the transport cycle. They have identified changes in the transporter in going from the outward to inward facing conformations, and upon substrate binding/release. The study is well laid out and thorough.

The author's results will prove quite informative and valuable to the transporter field. In this revision, the authors have also adequately addressed the issues raised in my review.

Thanks for the positive comments.

Reviewer #3 (Remarks to the Author):

The authors have made substantial revisions to the manuscript. A couple of remaining queries remain:

1. This reviewer previously inquired about signal sequences. The authors provided additional information about the possibility of a signal sequence in their response, but it is not evident that this has been added in the last paragraph of the manuscript as the authors indicate. It would also be appropriate to add Figure 3 of the response into the supplemental data with the associated text.

Thanks for the suggestion. The mentioned figure is combined with the previous Figure S1 in SI.

2. The authors agree that measuring cross-linked or variants for cysteine accessibility proteins function is important. However, the authors decided to measure only four of fifteen cysteine variants. The authors mention Q207, but this reviewer does not see that residue in Fig. 4 of the response. Is there functional data from a previous paper that could be described here. Neither is H536 shown. By focusing only on four residues, the authors examine only a subset. Just because residues are not conserved does not mean that they don't have a role. Therefore these cys residues and the experiments should be expanded beyond the four targeted in this initial re-submission.

Q207 is a residue which is variable among ZIPs. For instance, in human ZIP4, it is replaced by a histidine residue (H536). The function of H536 has been extensively investigated in our previous study (*Sci Adv*, 2017, 3, e1700344). When H536 was substituted by an alanine, it lost zinc transport activity by approximately 50-60% (fig. 3D in that report). As shown in fig.S12 in the same report, the K_M of the variant is about three times greater than that of the wild type protein. In fig.S15, the cell surface expression level of the H536A variant was similar or higher than that of the wild type protein. Collectively, the combined results indicated that H536 is neither essential for activity nor important for folding or trafficking. The reduced affinity, though, is consistent with the solved structure where it is involved in substrate binding in the primary transport site. To further clarify this point, the following sentence is added to the legend of Figure S8.

"The other residues subjected to cysteine substitution are either highly variable or have been functionally characterized in previous reports. For instance, the functional study of H536 in human ZIP4, which is topologically equivalent to Q207 in BbZIP, has been reported in Ref 46."

3. The authors raise an interesting issue with the problem of measuring protein function after mercury-crosslinking. Naturally, creating a Cys-less functional version of the transporter would ameliorate the issues the authors mention. Presumably, the authors do not want to attempt this. An alternative is not to use Hg cross-linking, but another cross-linking approach. Finally, is Hg transported by this protein?

We agree with Reviewer #3's comments about the challenge. Creating a cysteine-less ZIP4 variant is not trivial as it has 15 cysteine residues and many of them are conserved.

Organic bifunctional crosslinkers have a spacer between the two cys-reactive groups (maleimide), leading to an arm composed of at least nine atoms between the two sulfur atoms. This would result in a poorly locked conformation of BbZIP and accordingly it would be challenging to interpret the functional data of the crosslinked variant.

As of today, no ZIP has been reported to transport Hg^{2+} .

Reviewer #4 (Remarks to the Author):

The manuscript by Zhang et al., is a neat work that uses crystallography to obtain the apo-form of a bacterial ZIP metal transporter. The structure is resolved in the inward facing conformation (as the one resolved previously with the metal bound). They analyzed the possible dimerization of the ZIP transporter, suggesting that the dimer is unstable. Analyzing evolutionary couplings (and AlphaFold multimer) they suggest a dimer interface, validating it with cysteine crosslink. Finally, they generated an outward facing conformation using repeat swap homology modeling from the inward facing structure and validated this model with biochemistry assays. In this present form the work is interesting, the results seem robust, and it will certainly be of relevance in the field of membrane transporters. I have only few minor points.

1. The author should provide the details of the AlphaFold modeling. How many structures did they generate? Which database they used?

A section of "Structure prediction by AlphaFold" is added in Methods.

The following sentences are added to figure legend:

"predicted by AlphaFold, <https://alphafold.ebi.ac.uk/entry/Q6P5W5>" in Figure 9.

"All structures were retrieved from the AlphaFold Protein Structure Database (<https://alphafold.ebi.ac.uk/>)." in Figure S6.

Only one conformation was reported in the database or by AlphaFold Colab.

2. The authors may provide the full alignments for the repeat swap model or in the SI or in a database entry (i.e. Zenodo)

A new figure (Supplementary Fig. 11) showing the full sequence alignment is added in SI.

3. I do not agree with the statement that the repeat swap modeling has a bias toward the elevator mechanism. The main assumption of this approach is the definition of the repeats. Indeed, the same approach has been used to predict different mechanism, and these predictions depended only in the asymmetry between the repeats. Thus, the predicted outward facing model is another, not biased, evidence that supports the elevator mechanism.

We agree that the repeat swap homology modeling is a reliable approach in predicting alternative conformation. To address the concern raised by Reviewer #4, we altered the relevant sentence to the following one with new words being highlighted in yellow:

"exclude the potential artifacts that might be generated during modeling."